# Divergent dynamics of sexual and habitat isolation at the transition between stick insect populations and species

Patrik Nosil[1], Zachariah Gompert [2] & Daniel J. Funk [3]

Speciation is often viewed as a continuum along which populations diverge until they become reproductively-isolated species. However, such divergence may be heterogeneous, proceeding in fits and bursts, rather than being uniform and gradual. We show in *Timema* stick insects that one component of reproductive isolation evolves non-uniformly across this continuum, whereas another does not. Specifically, we use thousands of host-preference and mating trials to study habitat and sexual isolation among 42 pairs of taxa spanning a range of genomic differentiation and divergence time. We find that habitat isolation is uncoupled from genomic differentiation within species, but accumulates linearly with it between species. In contrast, sexual isolation accumulates linearly across the speciation continuum, and thus exhibits similar dynamics to morphological traits not implicated in reproductive isolation. The results show different evolutionary dynamics for different components of reproductive isolation and highlight a special relevance for species status in the process of speciation.

The formation of new species is a fundamental topic in biology. However, many questions regarding the speciation process remain unresolved[1–4], particularly concerning it's dynamics, i.e., how reproductive isolation changes over time[5–7]. (Here, reproductive isolation, RI hereafter, refers to inherent barriers to gene exchange, such as divergent mating preferences or low hybrid fitness[3,8–10]). This gap in our understanding reflects the challenge of studying a process that tends to occur over extended time periods. Thus, studies that focus on one or few taxa at particular points in the speciation process are unlikely to reveal how speciation dynamics unfold over time. This fact has motivated the development of the concept and study of the speciation continuum, the idea that we can observe and analyze an interconnected spectrum of closely-related populations and species varying in degree of differentiation and accompanying RI[11–14]. Although such comparisons do not allow a single speciation event to be studied from beginning to end[6], the hope is that by connecting patterns across taxa at different phases of divergence, one can retro-actively make inferences about the extended speciation process by getting an idea of the nature of the time course of speciation. Caveats here are that

speciation is likely complex, different taxon pairs may follow different trajectories to a given end-point, many taxa with partial RI may never proceed to evolve complete RI, and there can be many components of RI that exhibit different dynamics[6,15]. Nonetheless, as outlined below when covering past work on the speciation continuum, some insights can clearly be gleaned from such a comparative approach.

We stress that divergence across a continuum need not be uniform and gradual (Fig. 1). In fact, key questions remain concerning the conditions under which the speciation process is expected to be gradual versus more stepwise. For example, sudden transitions from weak to strong divergence might be driven by shifts in geographic distribution[3,16], population bottlenecks, the origin of large-effect mutations, or a non-linear snowball of genetic incompatibilities[17–19]. Theoretical work has highlighted how this can also happen if the effects of many weakly-selected loci become coupled (e.g., due to linkage disequilibrium) such that they reinforce the effects of selection on each other[6,20,21]. Related to these issues is the question of whether there are qualitative as well as quantitative differences between taxa at different phases of divergence: is species divergence simply

[1]CEFE, Univ Montpellier, CNRS, EPHE, IRD, Montpellier, France. [2]Department of Biology, Utah State University, Logan, UT, USA. [3]Department of Biological Sciences, Vanderbilt University, Nashville, TN, USA. e-mail: daniel.j.funk@vanderbilt.edu

## Hypotheses for the nature of the speciation continuum

### (A) Uniform strong relationship

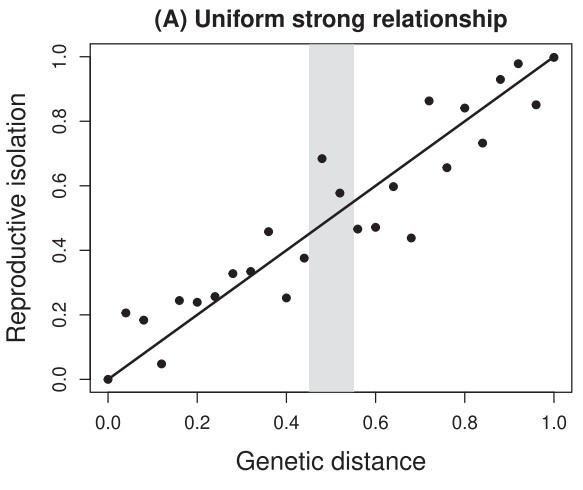

### (B) Uniform weak relationship

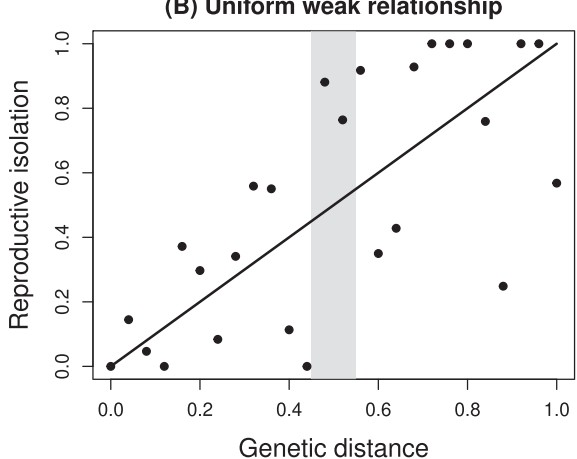

### (C) Heterogeneity associated with species boundary

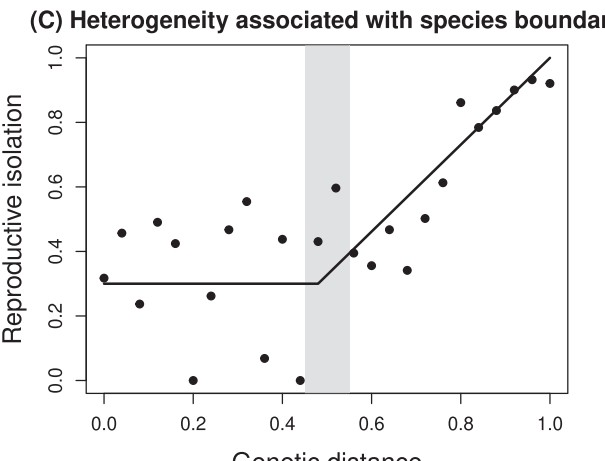

### (D) Heterogeneity independent of species boundary

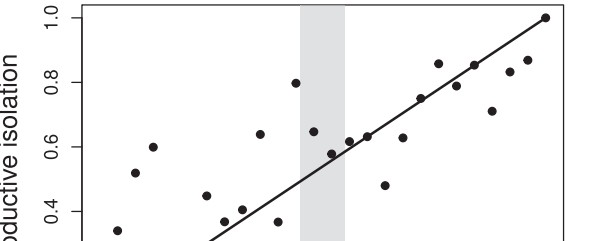

**Fig. 1 | Alternative hypotheses for the evolution of reproductive isolation as a function of genetic distance and the nature of the speciation continuum.** Here, species are defined based on taxonomic status. Points denote hypothetical pairs of populations and lines show the relationship between genetic distance and reproductive isolation. The shaded gray region denotes the species boundary. Reproductive isolation may evolve at a uniform rate with increasing genetic distance, and the relationship between these variables could be strong and relatively linear (**A**) or weak, with considerable variation among population pairs (**B**). Alternatively, reproductive isolation may only begin to accumulate with genetic distance after some minimal level of genetic divergence has occurred, with this transition either coinciding with the species boundary (**C**) or being independent of it (**D**).

population divergence writ large[22] or is the achievement of species status something more than just accumulating population differentiation? These questions have a long history in biology, with Darwin having emphasized gradual change[23], but the modern synthesis and subsequent genetic and theoretical studies having provided mechanisms for more abrupt increases in RI and completion of speciation (citations above). A focus on RI is justified given its central role in speciation (e.g., the Biological Species Concept equates speciation with the evolution of RI[9]) regardless of its evolutionary dynamics. Of course, as emphasized in a recent study by Anderson and Weir[24], evolutionary divergence of traits, including those responsible for RI, will occur regardless of their causal link to speciation as speciation progresses, and thus trait divergence and RI are not equivalent. We address these issues here using a combination of the two major approaches that have been used to study the speciation continuum.

First, classic studies quantified components of RI (e.g., ref. [25]) and tested their association with genetic distance, the latter generally assumed to reflect a proxy for time since divergence[1,26]. Typically, components of RI that could be assayed in the laboratory were studied, such as sexual isolation and intrinsic hybrid sterility and inviability. These studies, now implemented in flies (i.e., the classic study by Coyne and Orr[26]), frogs[27], fish[28], birds[29], butterflies[30], plants[31], and other

taxa, led to inferences about the rate at which different components of RI accumulate and the effect of geography on speciation. In most cases, these datasets focus on distinct and sometimes distantly-related species pairs such that information on the earliest, nascent phases of the process remained elusive. Moreover, the data used are often derived from many heterogeneous studies, rather than a focused, systematic examination of the evolution of a particular group. In such studies, time since divergence is usually not directly known and genetic distance is used as a proxy for time. However, demographic history can complicate the relationship between genetic distance and time. For example, periods of gene flow can erode genetic differentiation, leading to underestimates of divergence time, if this effect is not accounted for[32,33]. We thus here focus on genetic distance, analogous to the classic studies noted above, but also use demographic models to explicitly consider potential changes of interpretation in light of divergence time and gene flow.

Second, a group of recent studies has used modern genomic data to quantify genomic differentiation between taxa at different stages of divergence[5,6,34,35]. Although it is challenging to infer process from such genomic patterns alone[7,24,36,37], these studies have often revealed a continuum of differentiation, as observed in cichlids[38], pea aphids[14], stickleback fish[39], and mimetic butterflies[12]. Moreover, such studies

have also provided insights into the genomic basis of speciation. For example, studies of crows[40], sunflowers[41], and butterflies[42,43] have revealed evidence for highly heterogeneous patterns of genomic differentiation, with one or few marked peaks of genetic differentiation in isolated regions of the genome. Other studies, such as those in mosquitoes[44], cichlids[34], apple maggot flies[45,46], and stick insects[47–49] have demonstrated a combination of such peaks and more genome-wide differentiation (see ref. 35 for review). In contrast to the classic studies of components of RI noted above, these genomic studies have often focused on populations, ecotypes, or very recently-formed species pairs (which often still hybridize). Thus, evolution across the latter or final stages of the speciation continuum remains poorly quantified in such studies, some notable exceptions aside such as work in flycatchers[50,51]. Thus, collective past work has generally emphasized either the end or the beginning of the speciation continuum, rather than characterizing the entirety of the time course of speciation.

Perhaps most importantly, studies of genomic differentiation, which necessarily capture the consequences of total RI (along with other evolutionary processes and demographic factors), have rarely been integrated with experimental data to study how specific components of RI evolve as speciation unfolds (see ref. 5 for general review). Interestingly, such work is particularly called for given recent evidence that speciation rates in birds and flies are decoupled from rates of the evolution of RI[52]. Likewise, Price et al.[53] argued that the evolution of hybrid incompatibility is not the rate-limiting step in bird speciation, because this component of RI evolves more quickly than new niches become available for colonization. Our summary of past work above is not meant to be overly critical; many important insights have clearly emerged from these studies, as highlighted above. Rather, our goal is to pinpoint the nature of the work that remains to be done to better elucidate the dynamics of speciation. In this context, studies are required that span a wide range of differentiation across the speciation continuum in single radiations, and that integrate experimental estimates of components of RI with genomic data.

We offer such a study here, using 42 unique pairs of populations of *Timema*: wingless, herbivorous stick insects found throughout western North America[54,55]. These included conspecific population pairs collected on different host plants and between-species pairs, and cumulatively represented nine sexually-reproducing species obtained from California, USA (Fig. 2, details below). Specifically, we examine the evolution of two components of RI across the *Timema* speciation continuum. Our results provide evidence for both uniform and, most intriguingly, non-uniform evolution of RI as a function of increasing genomic differentiation, with implications for understanding the dynamics of speciation and the role played by species as evolutionary units.

Past genomic work on populations and species in this system documented a fairly uniform continuum of genomic differentiation, from weak to moderate to strong, when considering allopatric pairs of taxa[48]. A very different pattern was observed in sympatry. Here, only lowly (genome-wide $F_{ST} < 0.30$) differentiated populations and highly (genome-wide $F_{ST} > 0.70$) differentiated species were observed. Thus, intermediate levels of differentiation (genome-wide $F_{ST} > 0.30$ but < 0.70) were absent. In essence, there is a gap in levels of differentiation between the extremes, but this gap is restricted to sympatry. These collective patterns imply that populations that come into secondary contact below a threshold level of genomic differentiation rapidly collapse to lower levels of differentiation, for example, due to the homogenizing effect of gene flow that is known to occur within *Timema* species[48,56,57]. Thus, in sympatry one observes either weakly differentiated host-associated populations or strongly differentiated species, but not intermediate levels of differentiation. These results indicate a form of non-linearity in the dynamics of speciation, as predicted in some models of speciation[6,21,58]. The dynamics of components of RI across these phases of genomic differentiation remain

unclear, and form our focus here. We stress that it may be geographic isolation itself that allows genetic differentiation to build to the degree that taxa can be maintained as distinct units upon secondary contact[6,21]. Thus, our examination of the evolution of components of RI here considers aspects of demographic history such as divergence time and gene flow.

The current study thus differs from past work on the speciation continuum in the *Timema* system in four fundamental ways, which are critical for evaluating the dynamics of the evolution of components of RI. First, as described above, past work on the speciation continuum in *Timema* largely focused on genomic differentiation, not components of RI[48]. In that work, differentiation in sympatry is assumed to reflect total RI, but making inferences about RI between allopatric taxa using genetic differentiation is much more challenging[37,59], hence our emphasis here on components of RI inferred more directly from experimental data. Second, as described in more detail below, past work that considered trait evolution along the speciation continuum focused mostly on morphological traits that are not strongly associated with RI[48]. Such traits might exhibit different evolutionary dynamics during speciation than components of RI. Third, in past work that did consider components of RI directly, data was restricted to a single component of RI (sexual isolation) and for populations from within only a single species. Moreover, between-species variation was quantified largely between anciently-diverged non-sister species pairs whose evolution is not highly informative concerning the transition between populations and species (these data were collected for purposes other than studying this transition, see ref. 60). Thus, it remains unknown how components of RI evolve in *Timema* across the critical part of the speciation continuum where populations transition into species. Fourth, past work on divergent host-preferences specifically tested how preferences evolve as a function of shifts between different host plants in the context of the plants' evolutionary divergence[61–63]. This work used a much smaller data set than we provide here and did not examine host preference as a function of genetic distance; rather it focused on how divergence in host-plant use affects preference. Thus, we provide here an analysis of the role of host and mate preference in the speciation continuum, in the context of how preference varies with genomic differentiation.

Total RI can be comprised of contributions from many components of RI, of which we here study two likely common ones. Specifically, we estimate two components of RI between each of 42 taxon pairs of *Timema* (see Methods for details and Fig. 3, also Supplementary Data 1–4 and Supplementary Fig. 1 for details on populations, sample sizes, etc.). These 42 taxon pairs include many conspecific pairs that still exchange genes (see ref. 48 and estimates of gene flow in the current study). Consequently, because RI for many pairs is incomplete, these two components of RI did not entirely and always arise after speciation was complete but instead necessarily contribute to the ongoing speciation process[64]. The first component of RI was habitat isolation[3,16] due to divergent host-plant preferences. In insects that mate on their host plants, as do *Timema*, divergent host preferences can reduce contact and thus interbreeding between populations on different hosts, contributing to RI[65,66]. Divergent host-plant preference is likely to generate RI in *Timema* specifically because these insects feed, mate, and spend essentially their entire lives on their host-plants, and by virtue of being wingless they have limited dispersal[67,68]. In fact, mark-recapture studies have shown that many individuals spend their lives on the same single plant individual such that initial choice of that plant (which can occur, for example, when newly hatched nymphs emerge from eggs that overwinter in the soil) would negate mating possibilities with individuals on other plants[67]. The second component of RI was sexual isolation due to divergent mating preference. This is often considered an especially key component of speciation in the literature as genetic exchange always requires mating[3,69]. We note that we experimentally estimate two components of RI, not total RI.

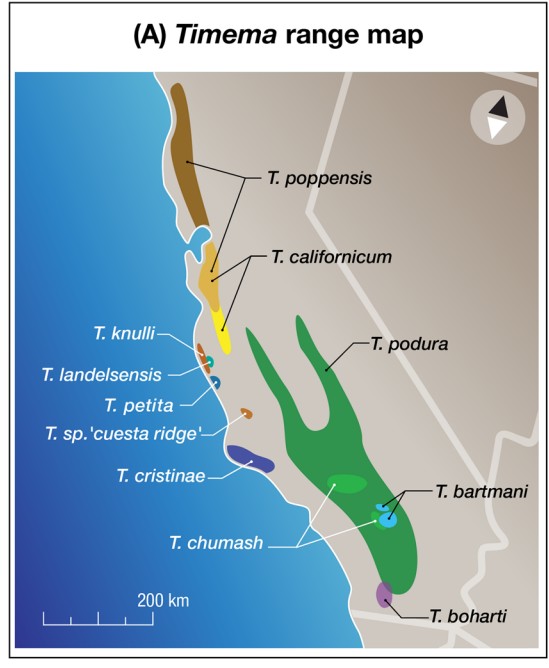

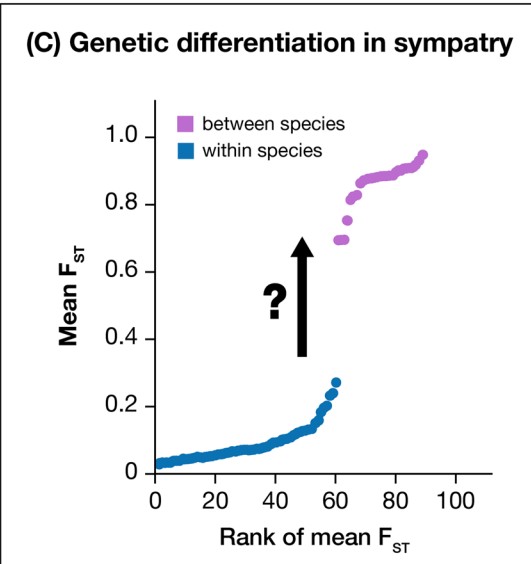

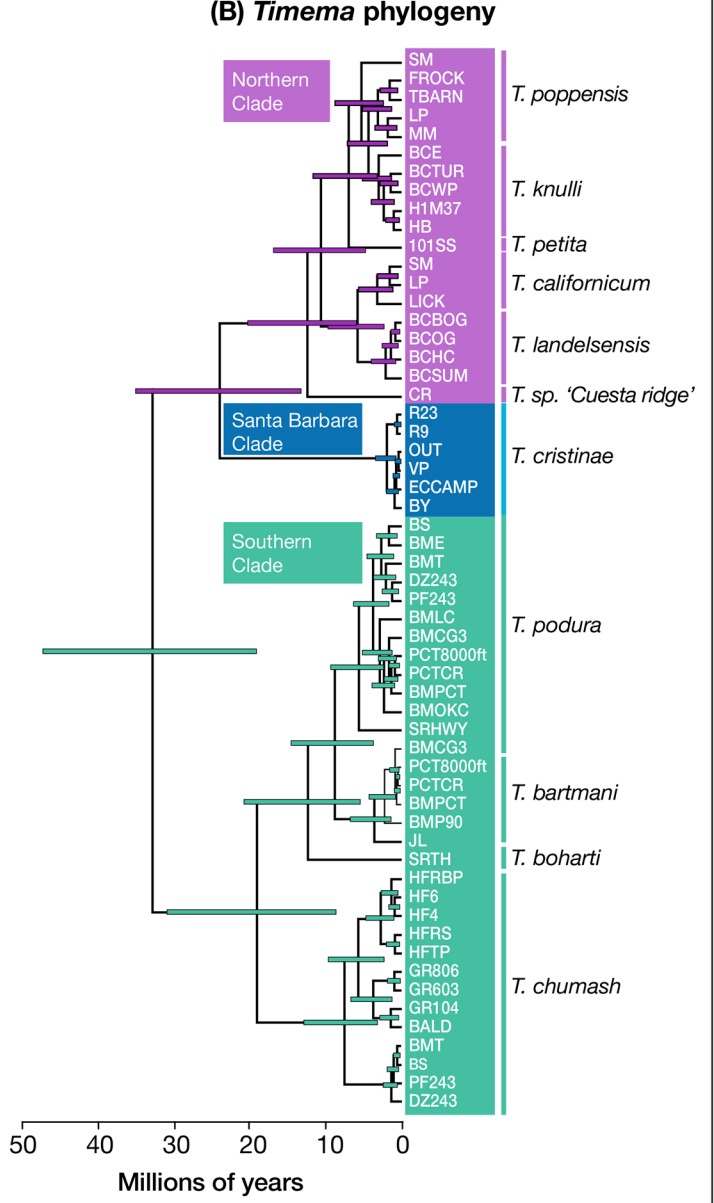

**Fig. 2 | Geographic distribution and evolution of *Timema* stick insects. A** Range map for sexual *Timema* species in California, USA. **B** Time-calibrated phylogenetic tree of the relationships between the *Timema* populations and species from ref. 48 (all nodes are supported with Bayesian posterior probabilities >0.97). **C** A gap in genome-wide $F_{ST}$ between conspecific populations and species pairs in sympatry based on genotyping-by-sequencing data (redrawn from ref. 48). A question mark denotes the unknown dynamics by which reproductive isolation evolves across this gap. Note that this gap was not observed when geographically-separated taxa (rather than sympatric pairs) were studied.

Moreover, our estimates come from experiments that do not fully recreate natural conditions (this is of course true of nearly all experimental studies of extrinsic RI). Thus, our experimental estimates of habitat and sexual isolation constitute proxies for how these components of RI would operate in nature. We nonetheless refer to these metrics as components of RI hereafter for simplicity and to be consistent with common usage in the literature (e.g., refs. 3,10,17,70).

We have previously shown that a suite of morphological traits—including body size and shape—diverge through time in a linear fashion[48]. These morphological traits do not clearly generate RI and thus act as a type of a priori expectation for evolutionary dynamics of traits that might be less critical for speciation. This allows us to here ask whether habitat and sexual isolation also exhibit linear dynamics similar to the morphological traits, or whether instead these

components of RI exhibit different (i.e., non-linear) evolutionary dynamics. Linear dynamics would not imply that habitat and sexual isolation are unimportant for speciation, as, after all, the evolution of RI is central to it under many verbal and formal models of speciation[3,9]. Moreover, we emphasize that we do not intend this as a test of the null hypothesis that components of RI exhibit different dynamics from other traits, as this is not the focus of this paper and would require a much larger suite of traits to generate a null distribution. Instead we use past work on these morphological traits as a baseline expectation for components of RI.

Our sampling design was specifically constructed to span a wide and fairly continuous range of genetic variation within-species and between closely-related species, using multiple different species at both levels of taxonomic status (e.g., multiple species for within-

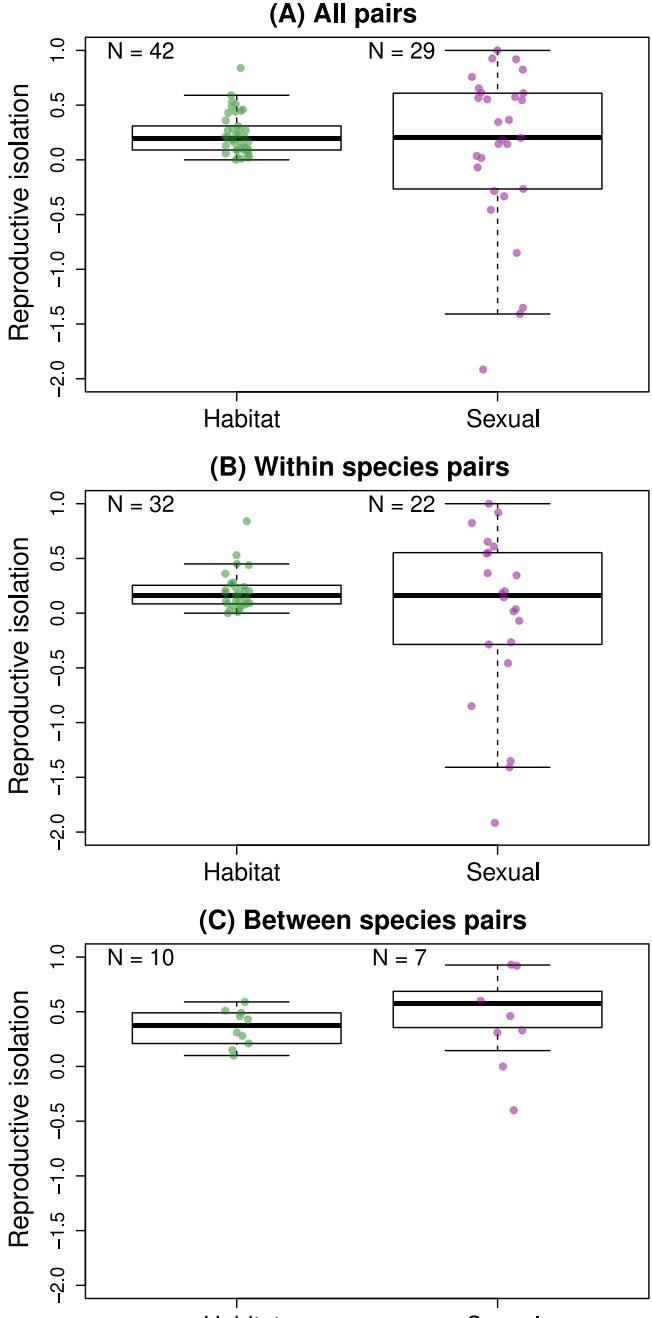

**Fig. 3 | Summary of habitat and sexual isolation in *Timema*, estimated using host-preference and mating preference trials, respectively.** Shown are results for all pairs of populations studied (**A**), only conspecific pairs, i.e., within-species variation (**B**), and only heterospecific pairs, i.e., between-species variation (**C**). Box plots show the median (midline), interquartile range (box) and range (whiskers, extended at maximum to 1.5 times the interquartile range) of habitat or sexual isolation, measured for pairs of taxa. Points denote estimates for individual population pairs. Sample sizes indicate the number of population pairs analyzed. Further details are enumerated in Supplementary Data 1 and 2. Source data are provided as a Source Data file.

species variation and multiple distinct species pairs). This allowed us to study the speciation continuum in a much more complete and systematic manner than is often possible. Most notably, the gap in $F_{ST}$ apparent in past work when considering only sympatric pairs (see Fig. 2) is bridged here by including allopatric and sympatric/parapatric population pairs. We estimated habitat isolation using 8749 host-

preference feeding trials involving all 42 pairs (mean trials per taxon pair = 208; Supplementary Data 1–3 for population-specific details). The same pair of host-plant species was used when testing each individual from a specific *Timema* taxon pair, with a wide range of the host plants used in nature represented among the different *Timema* taxon pairs, including both flowering plants (e.g., oaks, California lilac, chamise, manzanita) and conifers (e.g., pines and firs). Further details, scientific names, etc. can be found in the Methods and Supplementary Data 1–3. We estimated sexual isolation between 30 taxon pairs using 2436 mating trials (mean trials per taxon pair = 62; Supplementary Data 1–3 for population-specific details). These represented a large subset of the full 42 pairs, constrained by the fact that assaying sexual isolation is time-consuming and requires adult specimens obtained largely through labor-intensive rearing (see Methods for details). As noted above, we acknowledge that we here studied two components of RI (habitat and sexual isolation), among many possible ones. Future studies could usefully evaluate additional components of RI, including intrinsic postmating RI, to better understand the contribution of each to speciation. Nonetheless, as detailed above, we note that the two particular components of RI studied here are likely relevant for understanding speciation in *Timema* and mate preference is the component of RI studied in most classic studies of RI versus genetic distance[26,28–30]. Moreover, dynamics for these components of RI are compared to those for morphological traits thought to be less critical for speciation.

Additionally, we obtained estimates of nuclear (nucDNA) genetic divergence using 535 nucDNA sequences of 500 bp in length (anonymous nuclear gene Tc_nuc235; mean number of sequences per taxon pair = 15; Supplementary Data 1–3 for population-specific details) and combined these with analyses of published next-generation sequencing data obtained through genotyping-by-sequencing (GBS)[48]. The nuclear sequence data were critical as they allowed us to infer genetic distances for taxon pairs for which we did not have genomic data (due to the very strong correlation between the genetic distances inferred using the Sanger sequence and GBS data; see Fig. 4 and Methods for details).

Here, we document divergent dynamics in the evolution of habitat and sexual isolation within versus between species. Specifically, we show that habitat isolation is independent of genomic differentiation within species but accumulates linearly with it between species, whereas sexual isolation accumulates linearly within and between species. We emphasize the size and breadth of our dataset, involving a concerted effort to use many taxon-pairs and thousands of mating and host-preference trials to systematically quantify two components of RI across an unusually wide range of the speciation continuum, compared to previous studies, in a single group of closely-related taxa. Moreover, as the combined experimental and demographic results imply RI is not complete, the two components of RI studied here necessarily contribute to evolutionary divergence. Despite this concerted empirical effort, the limitations and caveats of this study are acknowledged in the Discussion.

## Results

### Habitat and sexual isolation vary widely within and between species

We observed a wide range of variability in the two components of RI we measured, with both habitat and sexual isolation ranging from absent to very strong, or even complete (Fig. 3, Supplementary Data 1). Notably, a wide range of variation in these components of RI was observed both among population pairs within species and between pairs of species. Thus, we strongly emphasize that the differences we report below within versus between species do not represent the lack of habitat isolation within species, but rather its independence from genetic distance within species. Genetic distance itself also varied fairly continuously across a considerable range (Fig. 4). Moreover, relative

## (A) Nuclear vs GBS

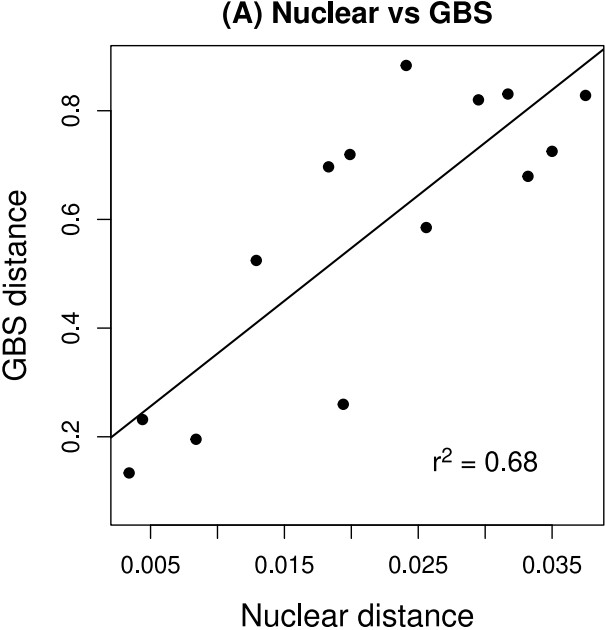

## (B) Genetic differentiation continuum

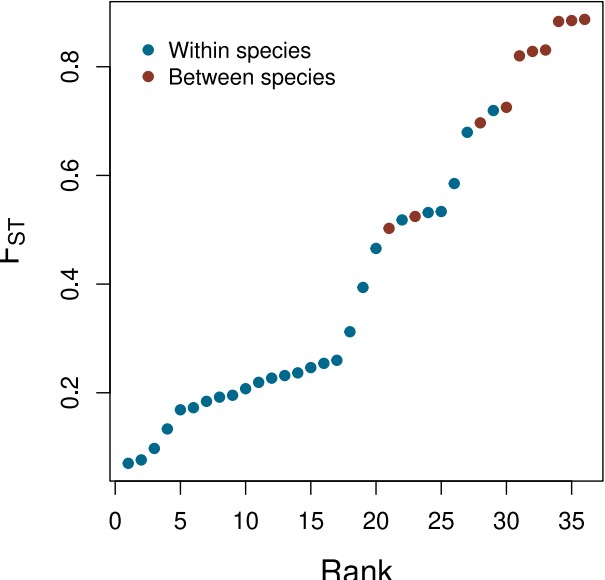

**Fig. 4 | Summary of genetic differentiation between pairs of populations and species. A** Relationship between DNA sequence divergence based on the anonymous nuclear gene (Tc_nuc235) and genome-wide $F_{ST}$ estimated from genotyping-by-sequencing data (GBS). Results are shown for $N = 14$ populations for which both data types were available. The best fit line from a simple linear regression is shown. **B** Rank sorted genetic differentiation ($F_{ST}$) for the $N = 36$ population pairs with genetic data used in this study. Pairs are colored to denote conspecific (within species, blue) versus heterospecific (between species, red) comparisons. The gap in $F_{ST}$ apparent in past work when considering only sympatric pairs (see Fig. 2) is bridged here by including allopatric population pairs. Source data are provided as a Source Data file.

genetic distances from GBS data ($F_{ST}$) were highly correlated with absolute distances based on DNA sequence divergence (Pearson correlation = 0.82, $r^2 = 0.68$, see Fig. 4A), indicating that our analyses are robust to relative versus absolute metrics of genetic distance (see below for explicit consideration of the effects of demographic history on genetic distances). Thus, the requisite variation for examining the evolution of RI across a wide range of genetic distances exists in our data set, and we did so with respect to habitat and sexual isolation, thereby characterizing these aspects of speciation dynamics in depth and at a fairly fine scale. Notably, sexual isolation and habitat isolation were not correlated with each other ($N = 30$ taxon pairs, Pearson $r = 0.171$, $P = 0.374$), foreshadowing the different dynamics for each reported below.

### Non-linear evolution of habitat isolation across the speciation continuum

We estimated habitat isolation via divergence between taxon-pairs in host-plant preference in behavioral preference assays (see Methods). To examine the evolution of habitat isolation (dependent variable) we fit Bayesian linear mixed models where habitat isolation could vary as a function of genetic distance, taxonomic status (i.e., population pairs within species or species pairs), and the interaction between genetic distance and taxonomic status (as in ref. 71). Preliminary analysis found no association between either component of isolation and geography, that is whether populations were allopatric or sympatric/parapatric: $r^2 = 0.028$ and $P = 0.29$ for habitat isolation, $r^2 = 0.079$ and $P = 0.14$ for sexual isolation; thus geographic setting was excluded from our further core analyses, but we consider future work on this in the Discussion. We then compared the fit of models with all these factors (full model hereafter) to those with subsets of the factors (various reduced models, including one with only the intercept) based on the deviance information criterion (DIC).

These analyses revealed that although habitat isolation varied both within– and between–species (e.g., Fig. 3), the manner in which it co-varied with genetic distance was different between these taxonomic levels (Figs. 5 and 6). Thus, the best supported model was the full model, with clear indication of an interaction between taxonomic status and genetic distance (Supplementary Data 4 and Supplementary Table 1 for statistical details). In this context, habitat isolation did not covary with genetic distance in within-species comparisons (i.e., the slope was not credibly different from zero, $\beta = -0.043$, 95% equal-tail probability interval [ETPI] $= -0.28$ to 0.19). In contrast, at the between-species level there was a clear positive association between habitat isolation and genetic distance (i.e., the interaction term was non-zero, $\beta = 0.27$, 95% ETPI = 0.074 to 0.46) (Fig. 5). We thus observed an overall non-linear or breakpoint-like pattern (see below) whereby a flat association between habitat isolation and genetic distance within-species transitioned to a steep positive association at the between-species boundary.

Further results indicate that these patterns noted above are robust to how species are delimited. For example, we conducted an analysis of habitat isolation versus genetic distance that did not use taxonomic status in any way. Specifically, we fit a breakpoint regression to the data while excluding taxonomic status (i.e., this analysis ignores taxonomic status). This analysis revealed a clear transition or break in the relationship between habitat isolation and genetic distance that corresponded with the taxonomic species boundary, although the latter was not used in the analysis in any way (Bayesian Information Criterion [BIC] breakpoint model versus no-breakpoint model $= -29.2$ and $-33.7$, respectively) (Fig. 5). In summary, the evolution of habitat isolation appears uncoupled from genetic distance within species, but accumulates with genetic distance at the between-species level, suggesting differences in the dynamics of this component of RI across different phases of the speciation continuum, and a possible special role for species in the evolution of RI (at least for habitat isolation).

### Linear evolution of sexual isolation across the speciation continuum

We estimated sexual isolation via divergence between taxon-pairs in mating preference in behavioral mating assays (see Methods). We

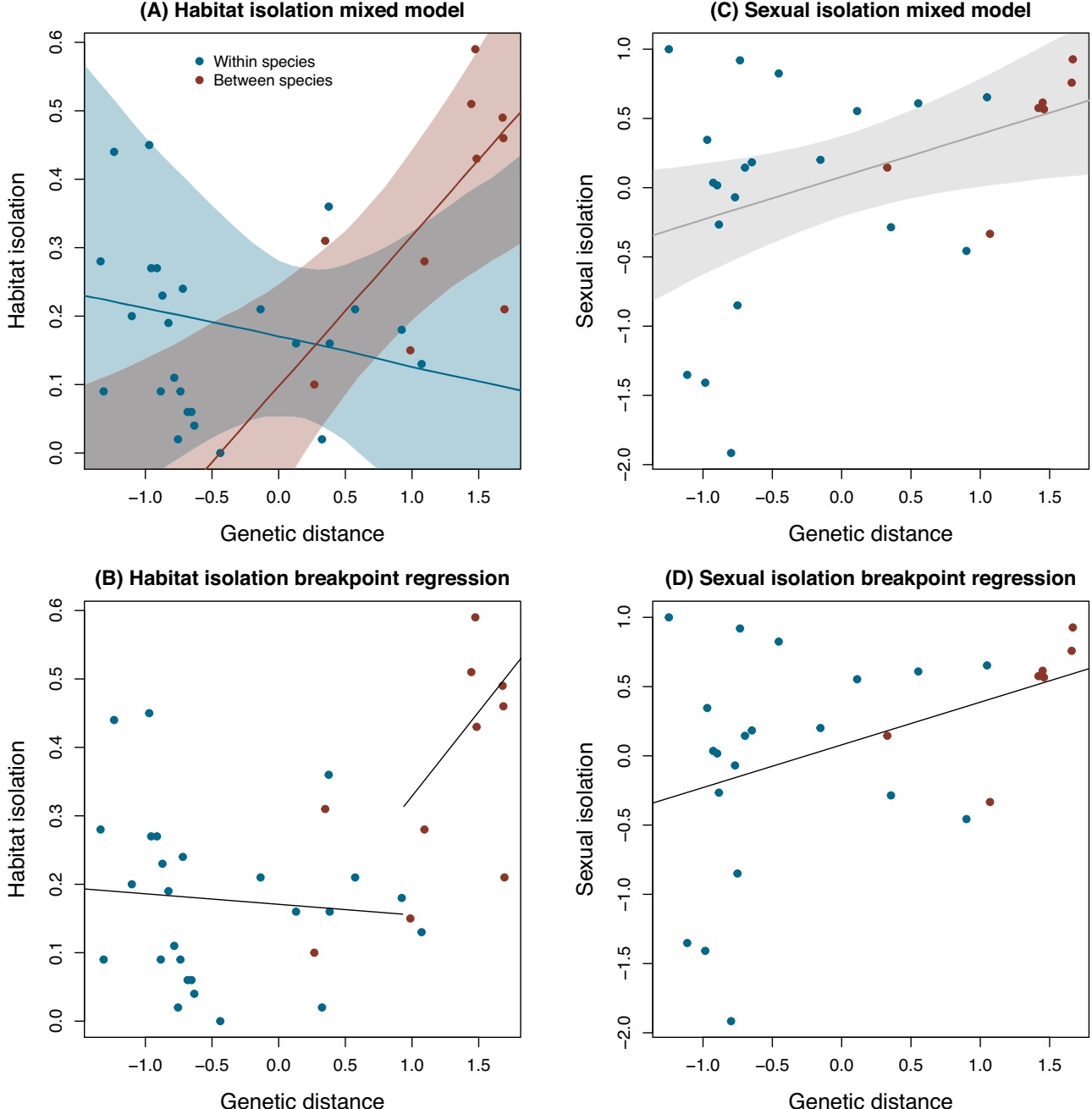

**Fig. 5 | Evolutionary dynamics of habitat and sexual isolation, where genetic differentiation is inferred by integrating GBS and Sanger sequencing data.**
**A** Shows the relationship between genetic distance ($F_{ST}$) standardized to have a mean of 0 and SD of 1 (this is $F_{ST}$–but a scaled version) and habitat isolation. Points denote $N = 36$ population pairs (colored by comparison type) with genetic and habitat isolation data. Lines and shaded areas denote Bayesian point estimates (posterior medians) and 90% equal-tail probability intervals (ETPIs), respectively. Results are shown for the best model (by Deviance Information Criterion [DIC]), which included an effect of genetic distance, comparison type, and an interaction term. **B** Shows the same scatterplot with best fit lines from a breakpoint regression

analysis ($N = 36$, Bayesian Information Criterion [BIC] without breakpoint = −29.2, BIC with breakpoint = −33.7). **C**, **D** Show similar results from the Bayesian model (**C**) and breakpoint regression analysis (**D**) with sexual isolation ($N = 27$ population pairs with genetic and sexual isolation data). The best model (based on DIC) for sexual isolation included only an effect of genetic distance. Thus a single best fit line and ETPI is denoted (**C**). Likewise, the breakpoint regression analysis favored a model without a breakpoint, and thus the simple linear fit without the breakpoint is shown in (**C**) ($N = 27$, BIC without breakpoint = 43.8, BIC with breakpoint = 37.7). Source data are provided as a Source Data file.

conducted the same analyses for sexual isolation as we did for habitat isolation above, but we observed different patterns (Figs. 5 and 6). For sexual isolation, the best model included genetic distance but not taxonomic status or an interaction term (Supplementary Data 4). Thus, under the best model, sexual isolation increased at a constant rate across the speciation continuum (i.e., genetic distance and sexual isolation were positively

correlated) ($\beta = 0.31$, 95% ETPI = −0.01 to 0.65, posterior probability $\beta > 0 = 0.97$) (Fig. 5). Moreover, even considering the full model (which received the least support based on DIC), there was no evidence of a credible effect of taxonomic status or a genetic distance by taxonomic status interaction on sexual isolation (Supplementary Tables 1 and 2). A supplementary Bayesian analysis that accounts for uncertainty in sexual isolation arising from finite sample sizes in

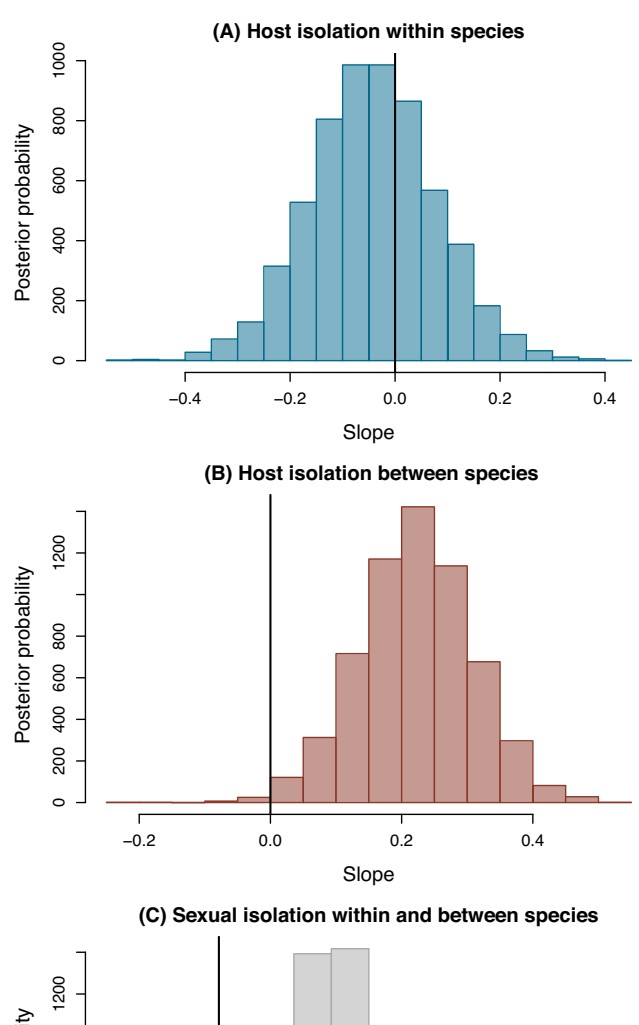

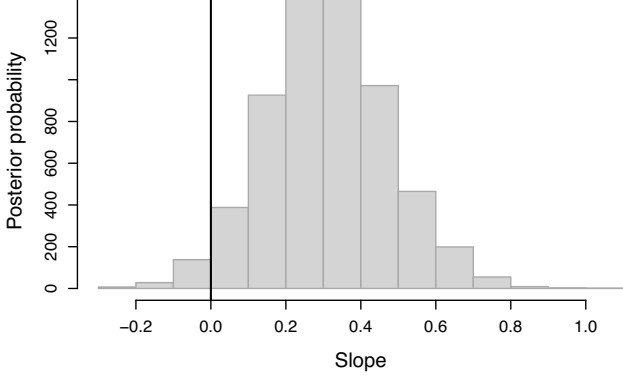

**Fig. 6 | Effect of genetic distance on habitat and sexual isolation within and between species.** Histograms show Bayesian posterior probability distributions for the effect of genetic distance on habitat within species (**A**) and between species (**B**), and sexual isolation within and between species pooled (**C**). Results are based on the full Bayesian linear mixed model with an effect of genetic distance, comparison type, and an interaction term for habitat isolation ($N = 36$ population pairs) and the model with only genetic distance for sexual isolation ($N = 27$ population pairs); these were the best models based on Deviance Information Criterion [DIC]. Vertical black lines denote the 0 value (slope of 0) within each posterior distribution. Source data are provided as a Source Data file.

mating trials gave similar results (see "Bayesian Model for Sexual Isolation with Uncertainty" in the Supplementary Methods for details). Consistent with these results, breakpoint regression analysis did not provide evidence of a breakpoint in the relationship between sexual isolation and genetic distance (BIC breakpoint model versus no-breakpoint model = 37.7 and 43.8, respectively) (Fig. 5). Thus, these collective results suggest that sexual isolation

exhibits linear evolutionary dynamics similar to morphological traits studied in the past (see Fig. 6 in ref. 48).

### Inference of divergence times and gene flow
Our core analyses used genetic distance (i.e., $F_{ST}$) as a proxy for time to study the dynamics of habitat and sexual isolation, similar to many classic studies (e.g., refs. 5,6,12,14,34). Moreover, genetic distance is especially informative in studies of speciation because genetic divergence not time per se ultimately creates new species. Nonetheless, gene flow and other demographic events, combined with the numerical properties of genetic distance measures, can influence the relationship between time and genetic distance. Thus, we used a combination of phylogenetics and historical demographic inference to quantify the relationship between divergence time and genetic distance ($F_{ST}$) in *Timema* to better understand the processes affecting the observed patterns of dynamics with respect to genetic distance (see Fig. 7, Supplementary Figs. 2 and 3, Supplementary Data 5 and Supplementary Tables 3, 4, and "Inferences of Divergence Time and Gene Flow" in the Supplementary Methods for detailed methods and quantitative results).

For the phylogenetic approach, we considered only population pairs where gene flow is expected to have a minimal influence on genetic differentiation. This includes between-species population pairs (past work has shown a lack of gene flow between *Timema* species[48,72]) and allopatric, conspecific population pairs (past work indicates little to no gene flow beyond about 15 km for *T. cristinae* populations[73]). Using divergence estimates for these pairs from a time-calibrated phylogeny[48], we found a strong but non-linear relationship between divergence time and $F_{ST}$ (model $r^2 = 0.78$, $P = 0.0002$) (Supplementary Table 3, Fig. 7A).

We complemented this phylogenetic approach by estimating divergence times under an isolation-with-migration model using $\delta a \delta i$ for all conspecific pairs with sufficient data[74,75] (i.e., unlike the phylogenetic approach, these models allow for gene flow). After excluding a single outlier that had an exceptionally high divergence time (about an order of magnitude higher than all others; see Supplementary Fig. 2A), our demographic estimates of divergence time were also strongly related to $F_{ST}$, but with some evidence of non-linearity (model $r^2 = 0.94$, $P = 0.0002$) (Fig. 7B and Supplementary Fig. 2B). We further compared this isolation-with-migration model to two alternatives, a strict isolation model and a model of isolation followed by secondary contact. The best model varied among population pairs (Supplementary Data 5 and Supplementary Table 4). When considering the best model for each pair, we failed to detect a significant relationship between divergence time and $F_{ST}$, though the trend was consistent with the results from the isolation-with-migration models. We discuss the implications of these findings in more detail below.

### Discussion
We analyzed here the evolution of two components of RI using a rich data-set that is well-replicated in both the number of experimental trials conducted and the number of taxon-pairs studied, and that spans the entire time course of speciation, representing one of the most concerted efforts to date to study the speciation continuum in a group of closely-related organisms. Our results have implications for understanding the dynamics of speciation and the nature of the speciation continuum. We discuss these in detail below, but first highlight key limitations and complexities in the interpretation of our results.

### General limitations and complexities in interpretation of the results
As acknowledged above, we studied two components of RI (habitat and sexual isolation), among many possible ones. Other components of RI, or even the same components of RI measured under different conditions, could exhibit different evolutionary dynamics, including dynamics that differ from traits not associated with RI. Future work

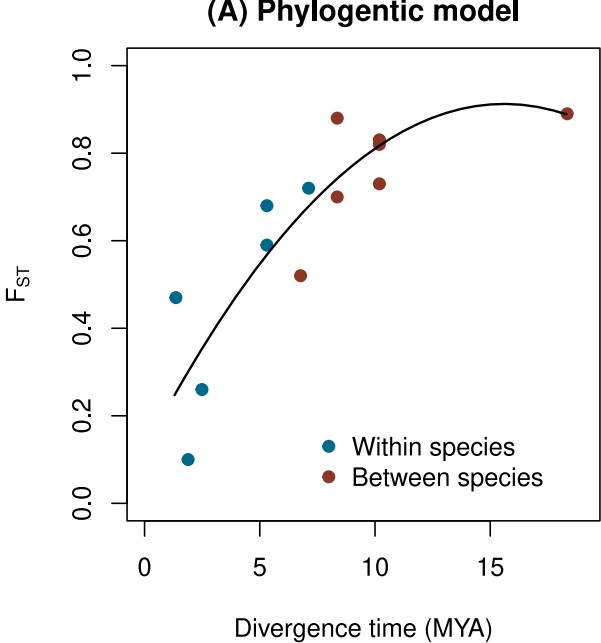

**Fig. 7 | Relationship between divergence time and genome-wide $F_{ST}$ estimated from genotyping-by-sequencing data (GBS). A** Shows $F_{ST}$ as a function of divergence time estimates (in millions of years ago, MYA) inferred from a time-calibrated phylogeny. Points denote each of 14 population pairs (see Supplementary Table 3) and are colored to indicate intra-specific versus inter-specific comparisons. The best fit line from polynomial regression is shown (model $r^2 = 0.78$, $N = 14$, $P = 0.0002$). **B** Shows $F_{ST}$ as a function of divergence time estimates (in units of $2 N_e$ generations) from a isolation-with-migration demographic model. Only intra-specific pairs were considered ($N = 9$ pairs; see Supplementary Data 5 and Supplementary Table 4). The best fit line from polynomial regression is shown (model $r^2 = 0.94$, $N = 9$, $P = 0.0002$). A single outlier with an exceptionally high divergence time was removed from this plot and model fit (see Supplementary Fig. 2). Source data are provided as a Source Data file.

could fruitfully examine additional components of RI, especially intrinsic postmating RI. Nonetheless, the two components studied here are very likely to generate meaningful RI in our study system and thus to be relevant for understanding speciation. This is especially likely as our demographic models suggest ongoing gene flow, which means that RI is incomplete between many taxon pairs and thus all components can make a contribution to the total RI.

Additional complexities arise from the partial decoupling of initial divergence time and $F_{ST}$ within *Timema* species suggested by the inferred demographic histories, which vary among taxon pairs and often include periods of gene flow. Thus, the relationship between genetic differentiation and each component of RI is relatively unambiguous, but interpreting the underlying patterns of genetic divergence and actual temporal dynamics by which the components of RI evolve requires more caution and nuance. This does not alter the dynamics we describe with respect to genetic distance (rather than time). Moreover, the generally strong relationship between $F_{ST}$ and time from the phylogenetic models and simpler isolation-with-migration demographic models suggests genetic distance is closely tied to time for many pairs of populations at various points along the speciation continuum.

**Dynamics of speciation and the nature of the speciation continuum**

Our results suggest that habitat isolation evolves non-uniformly across the speciation continuum in *Timema* stick insects. We again emphasize that the differences we found within versus between species do not represent the lack of habitat isolation within species, but rather its independence from genetic distance within species. In contrast, our results suggest uniform evolution of sexual isolation with genetic distance across the speciation continuum, and thus that sexual isolation exhibits similar dynamics to morphological traits not implicated in RI[48]. These findings build upon historical and modern debates on the

reality of species[3,16,76], and the extent to which species play a special role in evolutionary diversification[77]. If the speciation continuum is truly uniform, as is often currently thought, and as observed in some systems (e.g., *Heliconius* butterflies[12,78] and *Rhagoletis* flies[22]), then species divergence may simply be population divergence writ large. In such cases, the reality and evolutionary role of species can blur into those of populations. This appears to be the case for sexual isolation in *Timema*. In contrast, we report different dynamics of habitat isolation within versus between species, showing that species can be different in kind from populations, such that species status may play a critical role in the evolution of RI.

Concerning the general nature of the speciation continuum, an obvious question that emerges from our results is: Why does the observed difference in dynamics for habitat isolation within versus between species occur? The answer to this question likely involves, in large part, the effects of gene flow[3,16,79]. Note that we here consider gene flow mostly in the context of taxonomic status, which our study focused on, due to the reported lack of clear effects of geography on RI per se reported above. In this context, past work shows little to no current or historical gene flow between species of *Timema*, including those that currently co-occur in the same locality[48,80]. Thus, genomic differentiation between species likely largely reflects time since divergence (though not necessarily in a linear fashion), allowing for the fairly uniform accumulation of habitat isolation and sexual isolation with genomic differentiation between pairs of species in our study. This contrasts starkly with the situation within species, in which gene flow is known to occur as an increasing function of geographic proximity, leading to strong patterns of isolation-by-distance[48,56,57,80–83]. Here, genomic differentiation does not simply reflect time, but a complex interplay of population history, geography, gene flow, and even divergent selection which can counter gene flow[48,56,57,82]. It is this complex of evolutionary processes that likely explains our observed within-species uncoupling of habitat isolation

from genomic differentiation, calling for further study. We predict dynamics similar to those observed here will be observed in other systems where gene flow differs markedly at the within– versus between–species level. In contrast, in some other systems where gene flow and hybridization between species seems more common, such as butterflies[12,42,43,71,84], stickleback fish[39,85,86], and sunflowers[41,87], dynamics different from ours might be expected, perhaps yielding a more uniform continuum.

It may be particularly interesting to test how occasional historical introgression, as occurs in some butterflies[88] and cichlid fish[89], versus ongoing gene flow affect the evolution of the speciation continuum. Further studies in other systems that combine estimates of components of RI with genomic differentiation are clearly warranted to evaluate potential generalities and identify exceptions–and the mechanisms underlying each–in the speciation continuum and the associated time course of speciation. We suggest that studies comparing multiple components of RI among taxa that span the boundary between populations and species, as documented here, are particularly warranted. Indeed, the current dearth of comparable studies make it somewhat challenging to directly compare our results to past work, which has focused primarily on either early or very late stages of the speciation continuum.

Second, our results provide insight into the evolution of components of RI and their mechanisms, particularly in the context of the observed differences in patterns for habitat versus sexual isolation. Host preference and associated habitat isolation in phytophagous insects has played a particularly prominent role in debates concerning speciation with gene flow (e.g., sympatric speciation)[1,11,65,90,91]. This is due to the potential of divergent host preferences to reproductively isolate insect populations on different hosts (when such insects mate on their host plants), even in the same geographic area. In an analogous fashion, mating preferences have played a key role in discussions of reinforcement speciation[92,93]. In the *Timema* system specifically, host preference evolution does not appear to be radically different for populations that are geographically adjacent versus separate in *T. cristinae*, in which it has been studied[61]. In contrast, sexual isolation in *T. cristinae* is accentuated when host-associated populations are geographically-adjacent, in a reproductive character displacement fashion consistent with effects of reinforcement[94]. Although further studies are needed, we propose that our observed differences between components of RI could reflect the operation or dominance of different evolutionary processes affecting each, with reinforcement affecting mating preferences more strongly than host preference. Indeed, past work shows that host preferences in the *Timema* genus show marked evolution only when rare and extreme shifts occur between conifer and flowering-plant hosts, without a clear role for ongoing reinforcement-like processes[63]. Thus, different processes affecting each component of RI studied here could explain why the two components themselves were uncorrelated with each other.

Additionally, differences in the genetic basis of these components of RI might also contribute to the different patterns of evolution documented here (e.g., ref. 34). Both components of RI are likely at least partially heritable[48,61,62,94]. For example, host preferences of F1s between host ecotypes of *T. cristinae* are intermediate between the parental ecotypes[61]. Mating preferences do not appear strongly affected by host rearing environment and are based on polygenic variation in chemical communication traits (i.e, cuticular hydrocarbons)[48,60]. However, details of the genetic architecture of the components of RI studied here, and the traits underlying them, remain unknown. Thus, further work is required to test the effects of genetic architecture on the patterns of the evolution of RI documented here. Finally, we note that further work on components of RI that manifest themselves in a context-dependent manner in the wild, such as habitat preference and ecological selection against hybrids (e.g., refs. 95–97), are required to better contrast these components of RI with those that

manifest themselves strongly even in lab conditions. Furthermore, such contrasts will allow further evaluation of the often critical role of ecological speciation in diversification[1,2,10,70,86], and the extent to which total RI is under-estimated if inherently ecological barriers, such as habitat isolation, are not considered[98]. Indeed, had we not included habitat isolation in the current study, we would have missed the non-linear evolution of RI in this system and the consequent evidence for speciation being possibly distinct from population divergence writ large.

Third, our results inform the dynamics of speciation, showing that components of RI can evolve in a heterogeneous (across RI components) and non-uniform manner. Perhaps the most famous example of an expected non-linearity in speciation is the snowball effect for the evolution of intrinsic hybrid incompatibilities[17,19]. In this case, the number of gene-by-gene interactions that cause hybrid dysfunction accumulate in a non-linear fashion as the number of genes involved increases as each gene can potentially interact with multiple others, leading to an exponential rise in the number of potential incompatibilities. Here we provide evidence for another type of non-linearity, likely driven by the effects of gene flow, as discussed above. The collective results reinforce the fact that even divergence along a continuum can be uniform or non-uniform, and they challenge the notion that the speciation continuum is always truly a gradual and linear one[12,78]. Rather, speciation even along a continuum may proceed in heterogeneous fits and bursts. We also note that divergence along the continuum need not be uni-directional, especially if the dynamics involve homogenizing gene flow[38,99]. Divergence can be reversed, for example upon secondary contact, which can preclude simple relationships between genetic divergence and divergence time. Indeed, such a process might explain the absence of intermediate levels of genomic differentiation in *Timema* in sympatry[48].

Finally, our results have implications for species coexistence after speciation is complete. For example, they imply that a lack of gene flow between species prevents their collapse and fusion while at the same time facilitating the accumulation of RI and ecological differences that will make coexistence ever more possible. Indeed, *Timema* species whose ranges overlap tend to be anciently-diverged, and the only known example of species coexistence in the same locality on the same host-plant species involves two species that diverged many millions of years ago (*T. chumash* and *T. podura*)[48]. In these respects, our results on the non-uniform nature of the speciation continuum with respect to habitat isolation facilitate our understanding of speciation as a process and of post-speciation evolution as well.

In conclusion, we have shown that the speciation continuum can be heterogeneous both within and among different components of RI. The differences observed within versus between species for habitat isolation are consistent with a special role for species in evolutionary diversification, as originally envisioned during the Modern Evolutionary Synthesis (see refs. 3,16 for review), but as perhaps currently under-emphasized given accumulating genomic evidence for hybridization between species[5,78,100]. Further studies that combine estimates of RI with genomic differentiation have the potential to further inform the dynamics of speciation. We suspect that gene flow will play a key role here in explaining the patterns observed, and might help reconcile gradual views of evolution, as largely espoused by Darwin[23], with modern evidence that speciation can be a highly dynamic and heterogeneous process[18,19,21]. As such studies accumulate, we are poised to better understand not just patterns of speciation but the actual mechanisms underlying the origin of new species.

## Methods
### Sampling and animal maintenance
All specimens used in this study were collected using a sweep net, as in many past studies[101]. Briefly, this involves beating the branches of

host-plants, with a sweep net held underneath, and collecting the insects that fall down into the net.

As in past work, we define a population as all of the stick insects collected within a homogeneous patch of a single host species (see ref. 101 for review, refs. 48,63). Patches of two host species are sometimes distributed in interdigitated or adjacent patches that are in geographic contact with one another. We refer to insect populations associated with such patches as sympatric/parapatric. Other host patches are separated from patches of the alternative host by distances > 50 times the *Timema* 12 m per-generation migration distance[67] and are termed allopatric. This is a classification scheme which we have used in the past, but note that it is not critical here as geography is not the emphasis of our study. The host plant from which individuals were collected was recorded for each population at the generic level (Supplementary Data 1). Taxon pairs were chosen to span a likely continuum of divergence and thus both populations within species and species pairs are represented, but all within the major *Timema* clades to minimize phylogenetic effects and avoid the effects of ancient post-speciational divergence. The use of the same species in multiple comparisons was statistically accounted for in the Bayesian analyses as described below.

### Generation of host preference data and estimation of habitat isolation

Host preference was assayed using previously published protocols. We describe the main features of these assays here, and refer readers to past publications for further details[61–63]. Each evening, individual stick insects were placed in a 500 milliliter plastic cup with -15 cm cuttings of two different host-plant species. The cups were covered with mosquito mesh and left overnight. In the morning we scored which of the two plant species the stick insect was resting upon (trials where neither was chosen were excluded). Notably, *Timema* species are physiologically generalized in their diet, being able to grow and survive well on a range of host-plant species, including those never encountered in the wild or their likely evolutionary history[102]. Thus, choice of resting behavior is likely a good proxy for where *Timema* will mate and spend their lives, especially as dispersal over the bare ground of their chaparral habitats appears very rare[67,68]. These experiments tested field-caught individuals within a few days of collection. These individuals were a mixture of ages (i.e., sub-adult nymphs versus sexually-mature adults) as previous work in *T. cristinae* showed that age and even rearing in the lab on different hosts has little to no effect on host preference[61,62].

Habitat isolation was estimated as the degree of divergence between taxon pairs in host-plant feeding preference. This was estimated by assigning one of the two host species used in each assay as the reference host and then calculating the absolute value of: (proportion of trials in which taxon 1 of the pair picked the reference host) - (the proportion of trials in which taxon 2 of the pair picked the reference host). Note that the assignment of a reference host is totally arbitrary as the degree of divergence in preference is identical no matter which host species is picked as the reference. Full sample sizes and details are provided in Supplementary Data 1–3.

### Generation of mating data and estimation of sexual isolation

As for host preference above, mating preference was assayed using previously published protocols. We again briefly describe the salient features of these assays here, and refer readers to past publications for further details[94,103]. One adult male and one adult female were placed in a standard 10 cm petri dish. Each pair was observed for 1 h and scored as having mated or not (*Timema* have an extended copulation period that is unambiguous and easy to observe). Trials were conducted using the four mating combinations possible for each population pair (male

population 1 × female population 1, male population 2 × female population 2, male population 1 × female population 2, male population 2 × female population 1). The same protocol was applied to within- and between-species trials. These experiments largely tested field-caught individuals that were captured as virgin sub-adults and reared to sexual-maturity on the host upon which they were collected, in plastic containers where the sexes were kept separate to ensure virgin status of the individuals tested. Notably, previous work in *T. cristinae* showed that even rearing in the lab on different hosts has little to no effect on mate preference[94].

Using the results of these mating trials, we quantified sexual isolation as $1 - \frac{f_{het}}{f_{homo}}$, following ref. 26. Here, $f_{het}$ and $f_{homo}$ denote the frequency of matings for heterospecific (or between population) and homospecific (or within population) pairs, respectively. This index ranges from $-\infty$ to 1, with zero representing random mating, 1 representing perfectly assortative mating and thus complete sexual isolation and $-\infty$ representing perfect disassortative mating. This index was calculated for each taxon pair separately, using the four possible mating combinations described above. Full sample sizes and details are provided in Supplementary Data 1 and 2.

### Generation of DNA sequence data via Sanger sequencing and estimation of genetic distances

We obtained nuclear DNA sequence for an anonymous nuclear gene, Tc_nuc235 using primer pairs developed for *T. cristinae*: Tc_nuc235F = ATCCTGGAATTCACGCACTTAC and Tc_nuc235R = CTTACCCTTCTCCAAAATGTCG. PCR conditions[94,103]. DNA sequences were then obtained using Sanger sequencing, and trimmed and edited to retain 500 bps of high-quality data.

We estimated patristic distances (distances between tips in a phylogenetic tree) between sequences and then calculated the mean patristic distance between each taxon pair. Maximum-likelihood trees were reconstructed with PhyML version 3.0[104] using a K80+G for the nuclear dataset (best substitution models from jModeltest)[105]. The parameters of the substitution model, the alpha value of the gamma distribution, and the proportion of invariant sites were estimated by PhyML.

### Estimating genetic distances using combined genomic and Sanger data

As noted above, we measured DNA sequence divergence based on the anonymous nuclear gene Tc_nuc235 for 32 of the *Timema* populations pairs. Previously published estimates of genome-average genetic differentiation ($F_{ST}$) based on genotyping-by-sequencing (GBS) data were available for 19 pairs[48], including 14 of the pairs with nuclear DNA sequence data. Our goal was to obtain representative estimates of genome-average genetic differentiation ($F_{ST}$) for as many pairs of populations as possible. Genome-average $F_{ST}$ is a succinct but highly informative summary of genome divergence given the limited heterogeneity in $F_{ST}$ across the genome in *Timema*, especially outside of color and color pattern loci on chromosome 8[48]. To this end, we first fit a linear model for GBS-based $F_{ST}$ as a function of the nuclear gene DNA sequence divergence data for the subset of 14 pairs with both types of data. This was done in R (version 4.1.3). The linear model explained 68% of the variation in GBS $F_{ST}$ ($P = 0.0003$). Given the high association we detected between the two data sets, we used the regression coefficients from this linear model to predict GBS-based $F_{ST}$ for 17 population pairs with available nuclear DNA sequences but lacking GBS data. Specifically, we estimated GBS $F_{ST}$ as $0.15897 + 19.41020 x^{nuc}$, where $x^{nuc}$ was the nuclear gene DNA sequence divergence. This brought our total number of pairs of populations with GBS-based $F_{ST}$ to 36. We used this combination of observed and predicted GBS $F_{ST}$ estimates for subsequent analyses.

### Testing models for the accumulation of components of reproductive isolation along the speciation continuum

We fit Bayesian linear mixed models to determine how habitat and sexual isolation varied with genetic distance and whether this relationship differed for conspecific versus heterospecific population pairs. We did this using our estimates of $F_{ST}$ from the GBS data and the relationship between GBS $F_{ST}$ and nuclear DNA sequence divergence documented above. We used a mixed model approach to account for the fact that the same species (but never the same populations) were involved in multiple pairwise comparisons (analogous to refs. [71,106] where mixed models have similarly been used to account for non-independence of pairwise distance data). We specifically modeled RI (habitat or sexual isolation) as $RI_{ij} = \beta_0 + \beta_{gen}x_{ij}^{gen} + \beta_{taxon} + x_{ij}^{taxon} + \beta_{int}x_{ij}^{gen} * x_{ij}^{taxon} + \lambda_i + \lambda_j + \epsilon_{ij}$. Here, $x_{ij}^{gen}$ is the genetic distance (genome-average $F_{ST}$) for populations $i$ and $j$, $x_{ij}^{taxon}$ is a binary indicator variable that denotes whether the pair is conspecific (0) or heterospecific (1), the $\beta$s are estimated regression coefficients (including the intercept and interaction terms), the $\lambda$s are random effects denoting the average effect of pairs involving the species to which populations $i$ and $j$ belong, and $\epsilon_{ij}$ denotes the residual error. We placed relatively uninformative priors on the regression coefficients (normal with mean 0 and precision 0.001) and residual error variance (gamma with shape = 1 and rate = 0.01 for the reciprocal of the residual variance). We modeled the $\lambda$ random effects hierarchically, and specifically assumed a Normal prior on these parameters with a mean of 0 and a precision estimated from the data. We placed a gamma prior (shape = 1 and rate = 0.01) on the precision.

We centered and standardized the genetic distances before analysis. We then fit these models using the rjags interface with JAGS (version 4.13)[107]. Models were fit using Markov chain Monte Carlo (MCMC). We ran 2 chains for each model fit with 5000 iterations, a 1000 iteration burnin, and a thinning interval of 5. We calculated the Gelman-Rubin multivariate potential scale reduction factor with the coda package (version 0.19.4) to verify adequate mixing and likely convergence of the MCMC algorithm to the posterior distribution[108]. We repeated these analyses with submodels of our full model, including models without the interaction, with only genetic distance, with only the taxonomic effect, and a null model retaining only the intercept and random effects. Models were compared using deviance information criterion (DIC)[109].

### Breakpoint regression

We used breakpoint regression[110,111] as a complementary approach to determine whether the association between genetic distance and RI was fairly uniform or variable across the speciation continuum in a statistically significant manner. This was desirable as this analysis does not rely on taxonomic status in any way, thereby testing how the core results above are independent from how species are delimited. Breakpoint regression was conducted using the breakpoints function from the R package strucchange (version 1.5.3)[112,113]. Here, we modeled RI (habitat or sexual isolation) as a function of genetic distance ($F_{ST}$). We searched for a single, optimal breakpoint in the relationship between $F_{ST}$ and RI using a dynamic programming approach and with the constraint that each section must include at least 15% of the data. Models with and without a breakpoint were compared using Bayesian information criterion (BIC) and the residual sum of squares.

### Reporting summary

Further information on research design is available in the Nature Portfolio Reporting Summary linked to this article.

## Data availability

The reanalyzed DNA sequence data are available from the NCBI SRA database under accession PRJNA356405 https://www.ncbi.nlm.nih.gov/bioproject/356405). All other data are available from Dryad (https://doi.org/10.5061/dryad.q573n5tpk) and as a Source Data File associated with this manuscript. Source data are provided with this paper.

## Code availability

Core computer scripts used in the analysis of these data are available from GitHub (https://github.com/zgompert/TimemaRI) and Zenodo (https://doi.org/10.5281/zenodo.8312010)[114].

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

## Acknowledgements

We thank V. Soria-Carrasco for calculating the patristic distances and extracting the divergence time estimates and E. Janson, S.P.

Egan, S. Carpenter, D.P. Duran, S. Bai, W. Deacy, and C. Spear for help with data collection. This work was funded by a grant from the National Science Foundation of the United States of America (proposal number 0723379) to D.J.F. The support and resources from the Center for High Performance Computing at the University of Utah are gratefully acknowledged. This work was also supported by the French Laboratory of Excellence project "TULIP" (ANR-10-LABX-41) funded by a government grant as part of the France 2030 program, as a Senior Package to P.N.

## Author contributions

DJF and PN conceived the project. PN and members of the DJF lab collected data. ZG and PN analyzed the data. DJF provided funding. All authors wrote the manuscript and contributed to revisions.

## Competing interests

The authors declare no competing interests.
