## [Peer Review File · Nature Communications]

Divergent dynamics of sexual and habitat isolation at the transition between stick insect populations and speciesReviewers' Comments:

Reviewer #1:

Remarks to the Author:

This study follows on from previous studies on *Timema* speciation by measuring two different measures of reproductive isolation habitat isolation and sexual isolation.

For habitat isolation, the authors measure preference for host plant. They do this with a binary choice test, however, it is not clear how many different plants were used in the trials, or if the same plants/number of plants we used for each species pair comparison. This is important as the choices given to species likely have a strong influence on the choice test.

Additionally, I find the use of plant preference as a metric of reproductive isolation needs qualification. I understand these species live and mate on different plants, but is there any evidence that stick insects that prefer a particular plant are more likely to be found on that plant in nature? i.e. do these preferences actually relate to isolation?

Do the authors find that within species habitat isolation decreases with genetic distance or that there is no relationship? It is unclear to me.

For the second measure of RI the authors use sexual isolation. Surprisingly this shows no relationship to genetic distance either within species or even between species (with between species values being similar to within species), or when including geographical information. This is a really surprising result that is only briefly discussed by the authors despite the extensive work on sexual isolation previously found in this group.

Perhaps one reason is the coarse level of reproductive isolation used as a metric here? Looking at table S2 shows that many of the conditions examined have less than 10 replicates (while others have hundreds). This could give unreliable results when simply recording mating as a binary outcome. Happy to be wrong here but maybe it should be discussed?

Is there any correlation between habitat and sexual isolation? I would expect there to be.

Figure 2 – please add the sampled populations to the map. Please also provide the coordinates as a supplemental table as presently populations are just given as codes in the supplemental tables.

Figure 5. It would be nice to see this plot using F_{st} rather than a scaled genetic difference as the setup puts the question in terms of F_{st} (e.g. Fig 2C, lines 120-123).

The authors have not made their data and code available to me. It is recorded as 'pending' in their data statement so I have not been able to assess it.

Reviewer #2:

Remarks to the Author:

I have read the manuscript entitled "Divergent dynamics of reproductive isolation in stick insects at the transition between populations and species". While I appreciate the authors' efforts in investigating the dynamics of reproductive isolation during the speciation process, I have some concerns regarding the interpretation and scope of their results.

Firstly, I feel that the introduction could benefit from more explicit definitions and illustrations, particularly with regards to the authors' usage of the term 'reproductive isolation dynamics'. It would be helpful to clarify whether they are referring to the number of endogenous or exogenous barriers, their effects on one trait or several traits, or overall fitness. The concept of reproductive isolation is

sometimes a catch-all concept and it would be beneficial for the user of this concept to make explicit what they mean. Additionally, some statements made in the introduction require better justification, such as the interpretation of Figure 2-C, which is used to explain a non-linearity in the speciation process. This interpretation assumes a great deal of knowledge regarding *Timema* insects and their evolutionary history, which may not be familiar to all readers. In my case, which is neither familiar with *Timema* insects, nor familiar with reading into the *Fst* gaps, it does not seem obvious to me that the gap pointed out is the sole result of weakly selected mutations which at a "tipping point" would generate a strong reproductive isolation. I would naively say that it is the result of variations in demographic history among different populations/species, with cycles of geographic isolation and then secondary contacts. But I don't know what allows the authors to assume that all the divergence shown in this figure took place in sympatry. They must be clear on this point.

Furthermore, while I appreciate the authors' attempt to provide a general overview of the speciation process, the first five paragraphs of the introduction appear to be rather superficial and could be better balanced with more detailed information on the *Timema* insects, which are the primary focus of the study. The statement that speciation is a non-linear process is not novel (see works by Orr, Matute, Moyle, Morlon, Moritz, etc.) and therefore does not require such space. I believe that the introduction could benefit from a more focused approach to the main topic of the article.

In addition, I have some minor comments on specific statements made in the manuscript. For example, the term "speciation dynamics" is not well defined and could benefit from further clarification. Similarly, the suggestion that there is only one process of speciation ("the extended speciation process") is overly simplistic, as the biology of speciation is complex and influenced by a variety of factors.

Overall, while I appreciate the authors' effort in investigating the dynamics of reproductive isolation, I am concerned that the interpretation and scope of their results may be overreaching. In particular, the authors' study of two seemingly arbitrarily chosen traits (host plant preference and mating preference) may not necessarily be proxies for reproductive isolation, and the generalisation of their results to the dynamics of reproductive isolation accumulation may not be supported by the evidence presented. I must admit that I find it challenging to give them due significance while keeping in mind that there is no null expectation. Have the authors considered how the results might differ if we were to choose traits at random?

Therefore, while I understand the importance of the authors' work, I believe that the manuscript would benefit from further revision and clarification before publication.

Introduction

Major comments:

It may be beneficial to provide additional clarity and detail in the introduction (first five paragraphs) of the article. Specifically, more explicit definitions and illustrative examples would enhance understanding of the authors' intended meaning. In regards to the concept of 'reproductive isolation dynamics', it would be helpful to further elaborate on the specific types of barriers being considered, such as endogenous or exogenous, and their potential effects on various traits or overall fitness. The concept of RI dynamics thus seems to me to be interpretable in a very heterogeneous way according to the curricula of the different readers, and must therefore be explicitly defined here.

There are certain statements in the article that may require further justification to support their acceptance. Specifically, the interpretation of Figure 2-C that seeks to explain non-linearity in the speciation process could benefit from additional elaboration. Moreover, while the non-linearity of the speciation process is an interesting topic, it may not be as relevant to the main focus of the study as other aspects. Furthermore, the interpretation presented here assumes a significant level of familiarity with the literature on *Timema* insects, including hypotheses that may have been tested in past research but are not widely known among readers. While the pattern in Figure 2-C may suggest tipping points, there may be alternative explanations such as differences in demographic histories with periods of past allopatry before secondary contact. The authors must therefore recall here how these hypotheses have been ruled out in the past.

Continuing with the introduction, it can be distilled that speciation is a non-linear process. While this statement may hold varying interpretations for different readers based on their knowledge and background, it appears to have received a disproportionate amount of attention in the text. Conversely, the introductive part on *Timema* insects, which serves as the primary focus of this study, warrants more thorough elaboration. Consequently, it may be beneficial to re-evaluate the balance of the introduction to better highlight the central topic of the article which is not really "RI dynamics in the broad sense" but rather "host and reproductive preference in stick insects".

Minor comments:

Line 18: "how it's dynamics unfold over time".

What do the authors mean by "speciation dynamics" needs to be developed here?

Based on which criterion of species definition, as well as on which definition of reproductive isolation?

Speciation dynamics is an ongoing field of study that is receiving a lot of attention for theoretical, phylogenetic, population and experimental work. I therefore suggest that the authors clarify the questions explicitly addressed to better understand what they mean by "studying the dynamics of speciation".

Lines 25-27: "the hope is that ... time course of speciation".

This passage suggests that there is only one process of speciation, in one geographical and/or ecological context, under the influence of the same evolutionary forces acting with the same intensity. I do not feel that the biology of speciation can be reduced to the question of "where does the X-Y lineage pair fit into the process of speciation?"

Line 48: "although gene flow can complicate this interpretation of divergence time".

Clumsy formula: gene flow cannot be introduced here as only a noisy parameter.

Line 52: "genomic data to quantify genomic differentiation between taxa".

This is not exact, people are not only interested in "quantifying" F_{st} in nature, but in understanding the evolutionary processes behind the quantified F_{st} . The inference made is more interesting than the quantification itself.

Line 92: "which is known to occur within *Timema*".

It is imperative to provide more detailed information on a central point of *Timema* biology. To fully understand the dynamics at play, it would be beneficial to include a paragraph dedicated to the demographic history of the pairs under consideration, which can help clarify if the gene flow is due to secondary contact and if it is asymmetric. It is also important to provide estimated gene flow values. However, it is crucial not to oversimplify by reducing it to a mere statement such as "gene flow occurs".

Line 94: "thus, in sympatry one observes either weakly differentiated host-associated populations or strongly differentiated species, but not intermediate levels of differentiation. The results imply a non-linearity".

As an individual not extensively versed in the literature surrounding *Timema* insects, I may lack the necessary context to fully comprehend the significance of Figure 2-C. Instead of immediately attributing saltatory transitions from population to species, it would be beneficial for the authors to provide more background information regarding historical geography, in addition to current geography. Considering this, I wonder whether it is possible that the observed "gap" in F_{st} values could be attributed to previous geographic isolation prior to secondary contacts, rather than a tipping point scenario.

Line 95: "non-linearity in the dynamics of speciation".

The authors' use of the term "speciation dynamics" lacks a clear definition, which may lead to confusion for readers. It would be beneficial for the authors to provide more explicit details on what precisely they mean by "non-linear" in regards to speciation. For instance, do they refer to the accumulation of barriers or the effects of individual barriers on reproductive isolation, or perhaps the

total degree of reproductive isolation? Clarifying this point would greatly aid readers in understanding the authors' perspective.

Line 101: "past work on the speciation continuum focused on genomic differentiation, not RI". This sentence suggests a potential issue with prior research, as there may have been a tendency to transform descriptive studies on differentiation into investigations on a speciation continuum. However, it is unclear to me how previous authors addressed the concept of speciation without considering the role of reproductive isolation.

Line 106: "Thus, it remains unknown how RI evolves in *Timema* across the critical part of the speciation continuum where populations transition into species".

But at this stage it also remains unknown to the reader whether this critical part of the continuum would not correspond to past geographical isolation.

Line 116:

"causing reproductive isolation" or "contributing to reproductive isolation"?

Line 124: "the two forms of RI".

It may be helpful for the authors to provide a reminder of the potential connection between the two traits being investigated and overall reproductive isolation. Would random traits give the same results as those presented?

Results

Major comments

Since the study is primarily an analysis of the divergence dynamics of two traits in *Timema*, it seems central to demonstrate the relevance of these traits and in particular their role in reproductive isolation.

Why are these traits more important than other random traits?

What is the null distribution of trait divergence in *Timema*?

How are these traits more important than the dynamics of endogenous barrier accumulation?

In a recent paper by Anderson and Weir ("the role of divergent ecological adaptation during allopatric speciation in vertebrates"), the authors show that most of the traits suggested to be adaptive very often diverge in a Brownian manner between closely related species. It seems that the two traits considered here fall into this same pattern, and the observation reported here should be interpreted in the light of Anderson and Weir.

Minor comments

Line 142: "reproductive isolation varies widely".

To enhance clarity, I would suggest specifying the two variables directly measured rather than using the broad term "reproductive isolation".

Figure 3:

I propose to keep only one panel in:

1. deleting panel A, then placing the measurements made "within" and "between" side by side.

Line 146: "we strongly emphasize that the differences we report below within versus between species do not represent the lack of RI within species, but rather its independence from genetic distance".

Would this suggest that the two traits studied would not be relevant for studying the evolutionary dynamics of reproductive isolation?

Reviewer #1 (Remarks to the Author):

Point 1:

This study follows on from previous studies on *Timema* speciation by measuring two different measures of reproductive isolation habitat isolation and sexual isolation.

For habitat isolation, the authors measure preference for host plant. They do this with a binary choice test, however, it is not clear how many different plants were used in the trials, or if the same plants/number of plants were used for each species pair comparison. This is important as the choices given to species likely have a strong influence on the choice test.

Response 1: The reviewer is correct that host-plant species and sample size are important factors for interpreting the results. A complete list of this information was provided in Tables S1 and S2 (now Table S3) in the original submission, and the mean sample size was provided in the main text. We have retained this information in the revised manuscript. To supplement this information and thus further address the issue raised by the reviewer, we have now added a general summary of the plants used to the revised manuscript, as well as more detailed information on the sample sizes in the main text of the revised manuscript (lines 152-160).

Point 2:

Additionally, I find the use of plant preference as a metric of reproductive isolation needs qualification. I understand these species live and mate on different plants, but is there any evidence that stick insects that prefer a particular plant are more likely to be found on that plant in nature? i.e. do these preferences actually relate to isolation?

Response 2: As correctly noted by the reviewer, host-plant preference can translate into RI in this system because these insects feed, mate, and spend essentially their entire lives on their host plants. As requested, we have added further details to the revised manuscript that now allow better evaluation of the extent to which host preference is likely to translate to RI in nature. Particularly relevant details that we added concern the degree and life-history timing of dispersal and host-choice. Because these insects are wingless and move very little, and host choice occurs in newly hatched nymphs on the ground (where the eggs overwinter), host preference very likely generates RI in nature. These important life-history details have been added to lines 136-146 of the revised manuscript.

Point 3:

Do the authors find that within species habitat isolation decreases with genetic distance or that there is no relationship? It is unclear to me.

Response 3: We have clarified that there is no relationship; the association between preference and genetic distance is essentially flat. To ensure clarity on

this issue we have now provided our estimate of this effect in the revised manuscript (lines 208-210).

Point 4:

For the second measure of RI the authors use sexual isolation. Surprisingly this shows no relationship to genetic distance either within species or even between species (with between species values being similar to within species), or when including geographical information. This is a really surprising result that is only briefly discussed by the authors despite the extensive work on sexual isolation previously found in this group.

Perhaps one reason is the coarse level of reproductive isolation used as a metric here? Looking at table S2 shows that many of the conditions examined have less than 10 replicates (while others have hundreds). This could give unreliable results when simply recording mating as a binary outcome. Happy to be wrong here but maybe it should be discussed?

Response 4: Good point. We have added to the revised manuscript a new Bayesian analysis of the mating data that accounts for uncertainty arising from finite sample sizes. This analysis revealed that indeed there is a relationship between sexual isolation and genetic distance, supported both in a simple analysis using a new metric of sexual isolation and a new analysis accounting for uncertainty in the sexual isolation index. We thank the reviewer for leading us to make this important discovery. Details are reported on lines 227-240 and the new the “Bayesian Model for Sexual Isolation with Uncertainty” in the supplementary materials of the revised manuscript, and all figures and text have been updated accordingly. This new result does not affect the conclusions concerning non-linearity of habitat isolation, but adds a new result and richer context to the overall study.

Point 5:

Is there any correlation between habitat and sexual isolation? I would expect there to be.

Response 5: There is no correlation, which we now report in the revised Results section. Whether one is expected or not could depend on the processes driving the evolution of each form of RI, particularly the role of reinforcement-like processes in sympatry. We have added discussion of this issue to the revised manuscript (lines 192-193, 320-336).

Point 6:

Figure 2 – please add the sampled populations to the map. Please also provide the coordinates as a supplemental table as presently populations are just given as codes in the supplemental tables.

Response 6: We have added all this information to the Supplemental Materials (Table S2 and Figure S1), but opted not to add it directly to Figure 2 of the main

text because it resulted in a cluttered composite figure. We note in the revised manuscript where readers can find this detailed information in the Supplemental Materials.

Point 7:

Figure 5. It would be nice to see this plot using F_{st} rather than a scaled genetic difference as the setup puts the question in terms of F_{st} (e.g. Fig 2C, lines 120-123).

Response: This actually is a scaled F_{st} ; we have clarified this point in the revised figure and associated legend.

Point 8:

The authors have not made their data and code available to me. It is recorded as 'pending' in their data statement so I have not been able to assess it.

Response 8: As part of the revision process we have now made all our data and code available. As noted in the updated Data Availability Statement: The reanalyzed DNA sequence data are available from the NCBI SRA (<https://www.ncbi.nlm.nih.gov/bioproject/356405>). All other data are available from Dryad (doi:10.5061/dryad.q573n5tpk) (pre-publication access link: https://datadryad.org/stash/share/IVNYNroz44_GLvLq24OAbVkcYIzZwH32mPWrWwlwAfQ). Core computer scripts used in the analysis of these data are available from GitHub (<https://github.com/zgompert/TimemaRI>).

Reviewer #2 (Remarks to the Author):

Point 1:

I have read the manuscript entitled "Divergent dynamics of reproductive isolation in stick insects at the transition between populations and species". While I appreciate the authors' efforts in investigating the dynamics of reproductive isolation during the speciation process, I have some concerns regarding the interpretation and scope of their results.

Firstly, I feel that the introduction could benefit from more explicit definitions and illustrations, particularly with regards to the authors' usage of the term 'reproductive isolation dynamics'. It would be helpful to clarify whether they are referring to the number of endogenous or exogenous barriers, their effects on one trait or several traits, or overall fitness.

Response 1: We are pleased that the reviewer appreciates our data and results and have revised the manuscript to clarify the issues raised.

In particular, we have emphasized that by 'dynamics' we mean patterns over time. Practically, researchers may often not know time per se, and have used genetic distance as a proxy. As correctly noted below by the reviewer, gene flow and

other demographic events can influence the actual relationship between time and genetic distance. In this context, and in response to several comments below, we have added new demographic analyses of the genomic data that explicitly estimate divergence time and gene flow; this allows us to cast our results on RI in the specific context of divergence time, and genetic distance more generally as well (this is the classic approach, which we retain as it holds much information and because ultimately it is genetic divergence not time that causally creates new species). These revisions have resulted in a much clearer and more balanced manuscript, and we thank the reviewer for making us be clearer on these points.

We introduce these important issues in the Introduction of the revised manuscript

(lines 57-62) and our new demographic and divergence time results can be found in a new subsection of the revised manuscript entitled 'Demographic inference of divergence times and gene flow' (lines 241-271) of the revised manuscript. We also discuss the issue in the revised Discussion, Supplementary Materials, and a new main text figure (Figure 7) on the relation between F_{st} and divergence time.

Point 2:

The concept of reproductive isolation is sometimes a catch-all concept and it would be beneficial for the user of this concept to make explicit what they mean. Additionally, some statements made in the introduction require better justification, such as the interpretation of Figure 2-C, which is used to explain a non-linearity in the speciation process. This interpretation assumes a great deal of knowledge regarding *Timema* insects and their evolutionary history, which may not be familiar to all readers.

Response 2: Good points. We have revised the figure and main text to clarify that the concepts applied are general (lines 18-20, 29-32, 88-89, 138-146). At the same time, we have added detail on the study system (lines 138-146), and new results from demographic modeling (lines 241-271, Figure 7), to ensure the results are more readily interpretable.

Point 3:

In my case, which is neither familiar with *Timema* insects, nor familiar with reading into the F_{st} gaps, it does not seem obvious to me that the gap pointed out is the sole result of weakly selected mutations which at a "tipping point" would generate a strong reproductive isolation. I would naively say that it is the result of variations in demographic history among different populations/species, with cycles of geographic isolation and then secondary contacts. But I don't know what allows the authors to assume that all the divergence shown in this figure took place in sympatry. They must be clear on this point.

Response 3: Good point and we agree and absolutely did not mean to imply that divergence happened in sympatry. Rather, we agree with the reviewer that a likely scenario was divergence in allopatry with secondary contact - then leading either to genetic collapse through homogenizing gene flow or coexistence as divergent

species, depending on the level of differentiation upon secondary contact. In fact, this was our favored interpretation in past work and we definitely did not mean to imply here that divergence necessarily occurred in sympatry.

In this context, the idea of allopatry and a tipping point are not in conflict and in fact it may be allopatry that allows taxa to diverge to the point that they can coexist upon secondary contact. We have clarified these important conceptual points in the revised manuscript (lines 107-115). We discuss empirical aspects related to these concepts below, particularly in the context of new results from demographic modeling.

Point 4:

Furthermore, while I appreciate the authors' attempt to provide a general overview of the speciation process, the first five paragraphs of the introduction appear to be rather superficial and could be better balanced with more detailed information on the *Timema* insects, which are the primary focus of the study. The statement that speciation is a non-linear process is not novel (see works by Orr, Matute, Moyle, Morlon, Moritz, etc.) and therefore does not require such space. I believe that the introduction could benefit from a more focused approach to the main topic of the article.

Response 4: Although we agree that the idea of non-linearity in speciation will be familiar to some readers, it may not be to others, particularly at the broad readership of *Nature Communications* (we hope that some readers outside of speciation expertise will read the article if accepted). Thus, we have revised the Introduction to strike a balance between providing necessary but not superfluous background information.

Point 5:

In addition, I have some minor comments on specific statements made in the manuscript. For example, the term "speciation dynamics" is not well defined and could benefit from further clarification. Similarly, the suggestion that there is only one process of speciation ("the extended speciation process") is overly simplistic, as the biology of speciation is complex and influenced by a variety of factors.

Response 5: Agreed. As discussed in response to Point 1, we have clarified the meaning of any terms used and expanded our writing to embrace the potential complexity of the speciation process.

Point 6:

Overall, while I appreciate the authors' effort in investigating the dynamics of reproductive isolation, I am concerned that the interpretation and scope of their results may be overreaching. In particular, the authors' study of two seemingly arbitrarily chosen traits (host plant preference and mating preference) may not necessarily be proxies for reproductive isolation, and the generalisation of their results to the dynamics of reproductive isolation accumulation may not be supported by the evidence presented. I must admit that I find it challenging to give them due significance while keeping in mind

that there is no null expectation. Have the authors considered how the results might differ if we were to choose traits at random?

Therefore, while I understand the importance of the authors' work, I believe that the manuscript would benefit from further revision and clarification before publication.

Response 6: We appreciate the reviewers' point that not all traits may contribute equally to speciation and have explicitly acknowledged this point on lines 162-167 of the revised manuscript, citing the important study by Anderson and Weir. That said, we did not study any two totally 'randomly-chosen' traits. Rather, we directly estimated RI itself for two forms of RI that have a long history of consideration in speciation research. Host preference has long been considered of key importance in the classical literature on sympatric speciation (e.g., Bush 1969 in Evolution) and habitat preference more generally features strongly in modern discussions of eco-geographical isolation (e.g., Ramsey et al. 2003 in Evolution).

Likewise, mating isolation is considered a common and key component of speciation, and is the focus of numerous past theoretical and empirical studies, including the classical work of Coyne and Orr (1989 in Evolution) and bimodal hybrid zones (Jiggins and Mallet 2001 TREE). Thus, we openly acknowledge in the revised manuscript that our study is restricted to two forms of RI, but highlight how these are forms of RI itself (not arbitrary 'traits' such as body size or coloration) that are generally considered important for speciation (lines 136-146), and indeed we document some interesting patterns.

All that said, we fully acknowledge the reviewers point that many traits may or may not contribute to speciation and that comparing trait evolution to expectations under a null model is useful. However, this is different from studying and directly measuring RI in experiments as we do here. We have also cited all the studies noted above in the revised manuscript.

Point 7:

Introduction

Major comments:

It may be beneficial to provide additional clarity and detail in the introduction (first five paragraphs) of the article. Specifically, more explicit definitions and illustrative examples would enhance understanding of the authors' intended meaning. In regards to the concept of 'reproductive isolation dynamics', it would be helpful to further elaborate on the specific types of barriers being considered, such as endogenous or exogenous, and their potential effects on various traits or overall fitness. The concept of RI dynamics thus seems to me to be interpretable in a very heterogeneous way according to the curricula of the different readers, and must therefore be explicitly defined here.

Response 7: Agreed. As detailed in our response to Point 1 we have clarified these issues in the revised Introduction, alongside with adding more explicit

definitions. We thank the reviewer for encouraging greater clarity on these issues.

Point 8:

There are certain statements in the article that may require further justification to support their acceptance. Specifically, the interpretation of Figure 2-C that seeks to explain non-linearity in the speciation process could benefit from additional elaboration. Moreover, while the non-linearity of the speciation process is an interesting topic, it may not be as relevant to the main focus of the study as other aspects.

Furthermore, the interpretation presented here assumes a significant level of familiarity with the literature on Timema insects, including hypotheses that may have been tested in past research but are not widely known among readers. While the pattern in Figure 2-C may suggest tipping points, there may be alternative explanations such as differences in demographic histories with periods of past allopatry before secondary contact. The authors must therefore recall here how these hypotheses have been ruled out in the past.

Response 8: Agreed, and these conceptual issues have been considered as outlined in response to several points above.

In very concrete empirical terms, we have now conducted more explicit analyses of models of demographic history (applied to the genomic data in the manuscript) and added these new results to the revised manuscript. Together with past work on both historical introgression and contemporary admixture (e.g., our two papers in Nature EE in 2017 and 2022, both cited in context in the revised manuscript) these results clarify the demographic history of our study populations and also show that variability in such history does occur, but is unlikely to confound our results or their interpretations.

In sum, divergence generally does not appear recent and gene flow is high within- but not between-species. Most critically, there is a very strong correlation between F_{st} (genetic distance) and divergence time (estimated using both phylogenetic and demographic methods, that do not versus do account for gene flow, respectively). We report these new results in the aforementioned new subsection of the revised manuscript and new figures (Figures 7 and S2-S3, Tables S7-S9). Nonetheless, there is some nuance to our results which we also discuss.

We have presented these results in full and elaborated on them on lines 241-271 of the revised manuscript. More complex demographic modeling (i.e., that allows for multiple cycles of gene flow and isolation) is beyond the scope of the current study but is an interesting avenue for future research.

Point 9:

Continuing with the introduction, it can be distilled that speciation is a non-linear process. While this statement may hold varying interpretations for different readers based on their knowledge and background, it appears to have received a disproportionate amount of attention in the text. Conversely, the introductory part on *Timema* insects, which serves as the primary focus of this study, warrants more thorough elaboration. Consequently, it may be beneficial to re-evaluate the balance of the introduction to better highlight the central topic of the article which is not really “RI dynamics in the broad sense” but rather “host and reproductive preference in stick insects”.

Response 9: Good points. As noted above, we have re-written the Introduction to clarify the distinction between the forms of RI we measured and total RI, and added necessary detail to the Study System section.

Point 10:

Minor comments:

Line 18: “how it's dynamics unfold over time”.

What do the authors mean by “speciation dynamics” needs to be developed here?

Based on which criterion of species definition, as well as on which definition of reproductive isolation?

Speciation dynamics is an ongoing field of study that is receiving a lot of attention for theoretical, phylogenetic, population and experimental work. I therefore suggest that the authors clarify the questions explicitly addressed to better understand what they mean by “studying the dynamics of speciation”.

Response 10: Agreed that speciation is a conceptually rich field and that definitions need to be clear. We have clarified the meaning of all terms used and that our focus is on how the magnitude of specific forms of RI changes over time; that is how we use the term dynamics, consistent with past work on changes through time in the speciation literature (e.g., by Gavrilets, Flaxman, etc., citations added to the revised introduction) (lines 18-21).

Point 11:

Lines 25-27: “the hope is that time course of speciation”.

This passage suggests that there is only one process of speciation, in one geographical and/or ecological context, under the influence of the same evolutionary forces acting with the same intensity. I do not feel that the biology of speciation can be reduced to the question of “where does the X-Y lineage pair fit into the process of speciation?”

Response 11: Agreed. This statement has been modified and expanded to incorporate the potential complexity and plurality of speciation. We again thank the reviewer for forcing greater clarity on these issues (lines 29-32, 58-62).

Point 12:

Line 48: “although gene flow can complicate this interpretation of divergence time”.

Clumsy formula: gene flow cannot be introduced here as only a noisy parameter.

Response 12: Agreed. We have expanded and clarified this statement, noting that gene flow can erode genetic differentiation, resulting in under-estimates of divergence time, and acknowledged that the quantitative degree of gene flow matters (lines 58-62). Empirically, the new demographic analyses we have added to the revised manuscript address these issues explicitly.

Point 13:

Line 52: “genomic data to quantify genomic differentiation between taxa”.

This is not exact, people are not only interested in “quantifying” F_{st} in nature, but in understanding the evolutionary processes behind the quantified F_{st} . The inference made is more interesting than the quantification itself.

Response 13: Agreed. We have clarified that many processes can contribute to F_{st} , that it can be challenging to infer process from pattern (e.g., see lines 58-62 and 64-65 of the revised manuscript), and that process is of key interest to many scientists. We have empirically embraced this fact with our new demographic modeling results that report both gene flow and divergence time.

Point 14:

Line 92: “which is known to occur within *Timema*”.

It is imperative to provide more detailed information on a central point of *Timema* biology. To fully understand the dynamics at play, it would be beneficial to include a paragraph dedicated to the demographic history of the pairs under consideration, which can help clarify if the gene flow is due to secondary contact and if it is asymmetric. It is also important to provide estimated gene flow values. However, it is crucial not to oversimplify by reducing it to a mere statement such as “gene flow occurs”.

Response 14: Agreed. As noted, we present new results on demographic history and integrate them with past results on divergence time and gene flow. This has resulted in a clearer and more precise exposition of the history underlying the observed patterns of RI and genetic differentiation.

Point 15:

Line 94: “thus, in sympatry one observes either weakly differentiated host-associated populations or strongly differentiated species, but not intermediate levels of differentiation. The results imply a non-linearity”.

As an individual not extensively versed in the literature surrounding *Timema* insects, I may lack the necessary context to fully comprehend the significance of Figure 2-C. Instead of immediately attributing saltatory transitions from population to species, it would be beneficial for the authors to provide more background information regarding historical geography, in addition to current geography.

Considering this, I wonder whether it is possible that the observed “gap” in F_{st} values could be attributed to previous geographic isolation prior to secondary contacts, rather

than a tipping point scenario.

Response 15: As detailed in response 14 directly above and several other responses, we now provide explicit results on demographic history. Critically, we also outline how geographic isolation and tipping points are not mutually-exclusive alternatives; the former may allow the latter to be passed (as explicitly shown in Flaxman et al. 2014 Molecular Ecology, lines 111-114 of the revised manuscript). This critical point was not sufficiently clear in the original submission and the reviewer's comments have facilitated greater clarity on the issue.

Point 16:

Line 95: “non-linearity in the dynamics of speciation”.

The authors' use of the term “speciation dynamics” lacks a clear definition, which may lead to confusion for readers. It would be beneficial for the authors to provide more explicit details on what precisely they mean by “non-linear” in regards to speciation. For instance, do they refer to the accumulation of barriers or the effects of individual barriers on reproductive isolation, or perhaps the total degree of reproductive isolation?

Clarifying this point would greatly aid readers in understanding the authors' perspective.

Response 16: Agreed. We focus on how measured barriers build over time and with increased genetic divergence; the manuscript has been revised to clarify this (e.g., lines 18-21).

Point 17:

Line 101: “past work on the speciation continuum focused on genomic differentiation, not RI”.

This sentence suggests a potential issue with prior research, as there may have been a tendency to transform descriptive studies on differentiation into investigations on a speciation continuum. However, it is unclear to me how previous authors addressed the concept of speciation without considering the role of reproductive isolation.

Response 17: We have clarified what past work has done, and what was missing (lines 119-121 of the revised manuscript). Often, genetic differentiation for taxa that co-occur in the same locality is assumed to reflect RI. Making inferences about RI using genetic differentiation between geographically-isolated taxa is much more challenging. We have clarified these points in the revised manuscript.

Point 18:

Line 106: “Thus, it remains unknown how RI evolves in Timema across the critical part of the speciation continuum where populations transition into species”.

But at this stage it also remains unknown to the reader whether this critical part of the continuum would not correspond to past geographical isolation.

Response 18: Agreed, revised accordingly, particularly with respect to new results on demographic history.

Point 19:

Line 116:

“causing reproductive isolation” or “contributing to reproductive isolation”?

Response 19: ‘Contributing’ is fair; modified for clarity.

Point 20:

Line 124: “the two forms of RI”.

It may be helpful for the authors to provide a reminder of the potential connection between the two traits being investigated and overall reproductive isolation. Would random traits give the same results as those presented?

Response 20: Agreed, as noted above in response to Point 6:

“We appreciate the reviewers point that not all traits may contribute equally to speciation and have explicitly acknowledged this point on lines 162-167 of the revised manuscript. That said, we did not study any two totally randomly-chosen traits. Rather, we directly estimated RI itself for two forms of RI that have a long history of consideration in speciation research. Host preference has long been considered of key importance in the classical literature on sympatric speciation (e.g., Bush 1969 in Evolution) and habitat preference more generally features strongly in modern discussions of eco-geographical isolation (e.g., Ramsey et al. 2003 in Evolution). Likewise, mating isolation is considered a common and key component of speciation, and is the focus of numerous past theoretical and empirical studies, including the classical work of Coyne and Orr (1989 in Evolution) and bimodal hybrid zones (Jiggins and Mallet 2001 TREE). Thus, we openly acknowledge in the revised manuscript that our study is restricted to two forms of RI, but highlight how these are forms of RI itself (not arbitrary ‘traits’ such as body size or coloration) that are generally considered important for speciation (lines 136-146), and indeed we document some interesting patterns.

All that said, we fully acknowledge the reviewers point that many traits may or may not contribute to speciation and that comparing them in the context of null expectations is useful. We make this point and cite the associated paper in Science by Anderson and Weir in our revised manuscript (lines 162-167). We have also cited all the studies noted above in the revised manuscript.

Point 21:

Results

Major comments

Since the study is primarily an analysis of the divergence dynamics of two traits in *Timema*, it seems central to demonstrate the relevance of these traits and in particular their role in reproductive isolation.

Why are these traits more important than other random traits?

What is the null distribution of trait divergence in *Timema*?

How are these traits more important than the dynamics of endogenous barrier accumulation?

Response 21: Agreed, see response to related Point 6 for further details.

Point 22:

In a recent paper by Anderson and Weir (“the role of divergent ecological adaptation during allopatric speciation in vertebrates”), the authors show that most of the traits suggested to be adaptive very often diverge in a Brownian manner between closely related species. It seems that the two traits considered here fall into this same pattern, and the observation reported here should be interpreted in the light of Anderson and Weir.

Response 22: Agreed, and we have cited this important study in this context in our revised manuscript.

Point 23:

Minor comments

Line 142: “reproductive isolation varies widely”.

To enhance clarity, I would suggest specifying the two variables directly measured rather than using the broad term “reproductive isolation”.

Response 23: Modified as requested.

Point 24:

Figure 3:

I propose to keep only one panel in:

1. deleting panel A, then placing the measurements made “within” and “between” side by side.

Response 24: For clarity, we have retained the format as is.

Point 25:

Line 146: “we strongly emphasize that the differences we report below within versus between species do not represent the lack of RI within species, but rather its independence from genetic distance”

Would this suggest that the two traits studied would not be relevant for studying the evolutionary dynamics of reproductive isolation?

Response 25: As outlined in more detail in response to related Point 6, the ‘traits’ we studied are actually forms of RI and ones that are likely highly relevant in our system and that have received much historical and current attention in the speciation literature. That said, the point concerning the distribution of RI and total RI is well-taken and the issue is exposed in our revised manuscript with

reference to the study by Anderson and Weir (lines 162-167 of the revised manuscript).

Reviewers' Comments:

Reviewer #1:

Remarks to the Author:

The authors have addressed my concerns nicely and produced a cool paper.

The authors' data is also now accessible. My only comment would be that the code should be archived in Zenodo or similar before publication (rather than just as a GitHub repo which can be deleted...).

Reviewer #2:

Remarks to the Author:

After carefully revisiting manuscript NCOMMS-23-09353A, I am compelled to express my divided opinion on its contents.

1. On one hand, I must acknowledge the substantial improvement in the manuscript's writing. The inclusion of nuanced explanations and descriptive details pertaining to the biological model is commendable. It is worth highlighting the remarkable speed at which the demographic inferences were conducted, yielding results that align with expectations—an overwhelming prevalence of secondary contacts. This noteworthy outcome deserves publication as it enhances our comprehension of speciation within this group, examining the impact of major cycles of allopatric phases and secondary contacts on speciation. Consequently, it helps avoid the convoluted scenario of sympatric speciation involving specific traits.

2. On the other hand, this outcome challenges the relevance of studying two specific traits. With the presence of multiple secondary contacts, it seems more probable that reproductive isolation arises from endogenous barriers rather than divergence in the two aforementioned traits. Moreover, I maintain that an investigation focused on traits should include other traits as controls, devoid of any biased assumptions regarding their contribution to reproductive isolation. This approach would enable a comparison of the divergence dynamics between the two traits under scrutiny and the dynamics observed in random traits.

To summarize, in its current form, the manuscript's demographic inferences offer more intriguing results than the examination of trait dynamics. However, a comprehensive control study is imperative to accurately interpret the findings presented.

Two answers to answers.

Response 3: "We did not mean to imply that divergence happened in sympatry."

I respectfully hold a different perspective in response to the authors' statement. It may not be appropriate to simply acknowledge the reviewer's point by saying, "good point, we didn't mean...", as the original argument supporting the tipping point was based on the assumption of a sympatric context. If we were to consider an allopatric scenario, the interpretation of the F_{st} gap would be fundamentally different, rendering the concept of the tipping point irrelevant. In the previous version of the manuscript, the authors directly used tipping points to explain the F_{st} gap within the context of gene flow. Therefore, there exists a strong correlation between the interpretation of the "tipping point" in the initial version and the geographical context. Consequently, it appears contradictory for the authors to assert in their response to the reviewer that the "tipping point is not in conflict with allopatric divergence." In a secondary contact scenario, allopatry does not "allow" us to "reach the tipping point." Rather, allopatry facilitates the emergence of the observed F_{st} gap without necessitating the invocation of tipping points. Importantly, allopatry enables the accumulation of intrinsic barriers contributing to reproductive isolation, independent of specific traits.

Response 6: This point raises a significant concern that, regrettably, left me somewhat disappointed. My query was focused on the potential outcomes when considering random traits as a control group. In essence, it highlights the need for including additional traits to establish a comparative framework. While previous studies have indeed attributed importance to these specific traits in sympatric speciation, it is essential to question whether we should assign such significance to traits if their

divergence pattern mirrors that of randomly chosen traits (at least without any preconceived assumptions regarding their involvement in reproductive isolation). This underscores the criticality of incorporating control traits into the analysis. I apologize if my initial review was too subtle in highlighting this matter and would like to clarify it further. It appears challenging to draw meaningful conclusions without the inclusion of control measures that encompass "anonymous" traits and their potential contribution to reproductive isolation.

In my view, it seems implausible for a study examining trait dynamics to be published without adequate controls.

Reviewer #1 (Remarks to the Author):

Comment: The authors have addressed my concerns nicely and produced a cool paper.

Response: We are pleased to hear that the reviewer is satisfied with our revisions and finds the paper now suitable for publication. We thank them for their efforts in improving our work.

Comment: The authors' data is also now accessible. My only comment would be that the code should be archived in Zenodo or similar before publication (rather than just as a GitHub repo which can be deleted...).

Response: As requested, we have archived the code in a permanent repository, Zenodo (<https://doi.org/10.5281/zenodo.8312010>).

Reviewer #2 (Remarks to the Author):

Comment: After carefully revisiting manuscript NCOMMS-23-09353A, I am compelled to express my divided opinion on its contents.

1. On one hand, I must acknowledge the substantial improvement in the manuscript's writing. The inclusion of nuanced explanations and descriptive details pertaining to the biological model is commendable. It is worth highlighting the remarkable speed at which the demographic inferences were conducted, yielding results that align with expectations—an overwhelming prevalence of secondary contacts. This noteworthy outcome deserves publication as it enhances our comprehension of speciation within this group, examining the impact of major cycles of allopatric phases and secondary contacts on speciation. Consequently, it helps avoid the convoluted scenario of sympatric speciation involving specific traits.

Response: We are pleased to hear that the manuscript is improved and that the demographic models have enriched the study. It is true that we put great effort into conducting a substantial number of new demographic analyses in a timely manner for the initial revision. We are glad that this is appreciated and is deemed a strong addition to the manuscript.

Comment: 2. On the other hand, this outcome challenges the relevance of studying two specific traits. With the presence of multiple secondary contacts, it seems more probable that reproductive isolation arises from endogenous barriers rather than divergence in the two aforementioned traits. Moreover, I maintain that an investigation focused on traits should include other traits as controls, devoid of any biased assumptions regarding their contribution to reproductive isolation. This approach would enable a comparison of the divergence dynamics between the two traits under scrutiny and the dynamics observed in random traits.

Response: We empathize with the reviewers points and tried in the original revision to emphasize that although we do study two core 'traits' they are not 'arbitrary' - but rather represent components of RI inferred from experiments. Moreover, the components of RI studied here are ones with an enormous history of study in speciation research. Indeed, the classic Coyne and Orr type studies upon which our work is based focused on sexual isolation as one of the core components of RI important for speciation (along with endogenous barriers

as a second form of RI). We have tried to even better clarify these points in the further revised manuscript (e.g., lines 45-50, 152-153, 154-162, and 296-300 of the revised manuscript).

That said, we have gone further to address the issue of what traits 'matter' for speciation. In this context we have added discussion of two important issues to the revised manuscript.

1 - RI among populations within species is incomplete. In this context, any trait that generates RI contributes to reducing gene flow and to total RI, and thus to the active stages of speciation. By studying two forms of RI directly (not simply trait divergence or trait variation), including taxa that still exchange genes, these components of RI inherently matter for speciation. We did not simply study two arbitrary, for example morphological, traits - we study RI directly including among taxa that still exchange genes. In all due respect, we find it challenging to understand how RI itself is not of relevance for understanding speciation, for example, given its central role in the Biological Species Concept and many other verbal and formal models of speciation. We have made this point more clearly on lines 45-47, 142-143 and 300-302 of the revised manuscript.

2 - Control traits. We have further addressed this issue, as requested by the reviewer. For example, in past work we have shown that a suite of morphological traits - including size, shape, and coloration - diverge through time in a linear fashion (see Riesch et al. 2017 *Nature EE*, now cited in this context in the revised manuscript). These same traits do not clearly generate RI and thus they act as a type of a priori expectation for evolutionary dynamics of traits that might be less critical for speciation. In the revised manuscript, we have thus contrasted the evolution of the two explicit forms of RI we do study to this past work (see lines 10-11, 49-50, 124-127, 154-162, 183-184, 259-260, and 306-308 of the revised manuscript). And, of course, we continue to emphasize in the manuscript that the study of additional traits and forms of RI is warranted in future work, with now more explicit reference to endogenous barriers (see lines 296-298 of the revised manuscript). We hope that this satisfies the reviewers issue concerning what traits 'matter' for speciation.

Comment: To summarize, in its current form, the manuscript's demographic inferences offer more intriguing results than the examination of trait dynamics. However, a comprehensive control study is imperative to accurately interpret the findings presented.

Response: We agree the demographic inference enriches the study. As discussed in the response directly above, we have clarified the importance of studying RI directly, and contrasted the dynamics of these forms of RI to other traits.

Comment: Two answers to answers.

Response 3: "We did not mean to imply that divergence happened in sympatry."

I respectfully hold a different perspective in response to the authors' statement. It may not be appropriate to simply acknowledge the reviewer's point by saying, "good point, we didn't mean...", as the original argument supporting the tipping point was based on the assumption of a sympatric context. If we were to consider an allopatric scenario, the interpretation of the F_{st} gap would be fundamentally different, rendering the concept of the tipping point irrelevant. In the previous version of the manuscript, the authors directly used tipping points to explain the F_{st} gap within the context of gene flow. Therefore, there exists a strong correlation between the interpretation of the "tipping point" in the initial version and the geographical context. Consequently, it appears contradictory for the authors to assert in their response to

the reviewer that the "tipping point is not in conflict with allopatric divergence." In a secondary contact scenario, allopatry does not "allow" us to "reach the tipping point." Rather, allopatry facilitates the emergence of the observed *F_{st}* gap without necessitating the invocation of tipping points. Importantly, allopatry enables the accumulation of intrinsic barriers contributing to reproductive isolation, independent of specific traits.

Response: We again sympathize with the reviewers response here, as we did with the original comments on tipping points. However, it is simply factually inaccurate to state that the tipping point argument was 'based on the assumption of a sympatric context'. We apologize for the extensive text below, and we don't want to belabor the point, because as we outline below we have removed reference to tipping points in the revised manuscript as a further concession to the reviewer. However, we do want to illustrate that our initial response was not simply for 'convenience' and that past work on tipping points explicitly allows for, and explored, the role of allopatry, and the maintenance of differentiation on secondary contact.

For example, quoting from our 2017 empirical paper on the *F_{st}* gap (Riesch et al. *Nature EE*), emphasis added here in bold:

"The documented gap between sympatric ecotypes and species thus likely reflects intraspecific gene flow (i.e., incomplete RI) that prevents maximal differentiation from forming or **being maintained** in sympatry. In principle, the gap could be due to rapid sympatric speciation. **However, this is difficult theoretically^{2,18} and it does not match biogeographic patterns in *Timema***, where range overlap between taxonomically recognised species is slight or absent³⁶. Our results suggest that gene flow can contribute to evolutionary gaps. Specifically, gene flow can make intermediate phases of speciation difficult to observe because these phases occur rapidly (**e.g., in reverse**), rarely, or restricted in space. In such cases, gaps are 'apparent' rather than real and extensive sampling is required to observe intermediate states."

From our 2017 Perspective on tipping points (Nosil et al. *Nature EE*), emphasis added here in bold:

"Sudden speciation in the aforementioned model occurred without intrinsic genetic incompatibilities, major effect loci, genome re-arrangements, **or periods of geographic isolation, though these factors can promote the process**. For example, divergence maintained despite migration was often higher when initial differentiation **involved a period of geographic isolation** than when it did not⁵. This provides one example of bi-stability; two outcomes were possible for the same selection strength, **dependent on initial geographic setting**. This example also highlights that the divergence process is bi-directional, as differentiation can build, **be maintained upon secondary contact**, or collapse. Here, we propose a framework for understanding these potentially complex dynamics that draw parallels between speciation and tipping points in other complex systems⁶⁵⁻⁶⁷."

And from the original 2014 publication on the underlying model (Flaxman et al. *Molecular Ecology*), emphasis added here in bold:

"Periods of allopatry, chromosomal linkage among loci, and large-effect alleles can facilitate this process under some conditions, but are not required for it."

“With respect to allopatry, two predictions follow from GWC. First, when divergence in allopatry is **followed by hybridization upon secondary contact**, GWC predicts that populations should either quickly fuse or remain isolated; intermediate levels of mixing should be comparatively rare because populations are predicted to either be undifferentiated or highly divergent, (Fig. 3B; Fig. S7, Supporting information). In empirical cases **when secondary contact leads to intermediate levels of mixing**, we predict that genes of large effect may therefore often underlie divergence, as they would be less prone to homogenization (Fig. 3A; Fig. S6, Supporting information; see also Discussion). Despite homogenization of small-effect variants, **the period of allopatry may still be important** for incrementally increasing the chances for speciation in sympatry compared with primary contact scenarios by infusing populations with standing variation above baseline levels resulting from drift (albeit at very similar frequencies between populations). A second prediction about allopatry is that when sufficient standing variation for GWC exists, **a period of geographical isolation**, even without additional mutations, should elevate LD and thereby help trigger GWC, which would keep populations diverged even if migration resumed. Simulations supported this prediction as well (Figs 3B and 4B; Fig. S7C, Supporting information).”

Thus, our initial revisions were not simply conducted in a manner that was ‘convenient’ for the response, but accurately reflect past work.

That said, we do want to avoid any confusion and desire to satisfy the reviewer in this regard. To further address the reviewer’s concerns, we have deleted all reference to tipping points in the further revised manuscript. Instead we focus on the demographic context of divergence, based on the modeling we present. We hope that this concession further satisfies the reviewer.

Comment: Response 6: This point raises a significant concern that, regrettably, left me somewhat disappointed. My query was focused on the potential outcomes when considering random traits as a control group. In essence, it highlights the need for including additional traits to establish a comparative framework.

While previous studies have indeed attributed importance to these specific traits in sympatric speciation, it is essential to question whether we should assign such significance to traits if their divergence pattern mirrors that of randomly chosen traits (at least without any preconceived assumptions regarding their involvement in reproductive isolation). This underscores the criticality of incorporating control traits into the analysis. I apologize if my initial review was too subtle in highlighting this matter and would like to clarify it further. It appears challenging to draw meaningful conclusions without the inclusion of control measures that encompass “anonymous” traits and their potential contribution to reproductive isolation. In my view, it seems implausible for a study examining trait dynamics to be published without adequate controls.

Response: Please see response above, in terms of studying RI directly, particularly between taxa that still exchange genes, and comparison to the evolution of other traits. These collective changes have further clarified the ideas and results in the manuscript and we thank the reviewer for facilitating these revisions.

Reviewers' Comments:

Reviewer #2:

Remarks to the Author:

Having carefully reviewed the most recent iteration of the manuscript titled "Dynamics of Reproductive Isolation in Stick Insects at the Transition Between Populations and Species," I must candidly express that my reservations remain unaddressed by the authors.

The issue of experimental control persists without resolution. Mere citation of another article, wherein the relevant information is similarly obscure, falls short of constituting a control.

The question pursued by the authors is undeniably intriguing and holds paramount significance within the field of speciation biology - the comprehension of the evolutionary dynamics underpinning reproductive isolation. Nevertheless, my skepticism regarding the appropriateness of relabeling isolated traits as "reproductive isolation" remains unabated. This semantic shift essentially transforms the conducted experiments into an inquiry into reproductive isolation, a conceptual realignment that I consider fundamentally misguided. The assertion that other authors employ a similar approach lacks compelling substantiation. Reproductive isolation encompasses more than the two traits under scrutiny; genomes abound with various incompatibilities contributing to reproductive isolation.

Within this framework, wherein the quantified traits represent but an enigmatic facet of reproductive isolation, the absence of a discernible correlation between the "divergence of these traits" and "genetic divergence" does not constitute a result worthy of publication. Do these traits contribute to reproductive isolation with uniform potency across all pairs? Did these traits evolve before or after the emergence of other barriers for all pairs?

The discussion concerning demographic history remains conspicuously lacking in depth. Despite the execution of demographic analyses, the findings present a confluence of diverse scenarios, some involving isolation and others not, further complicating the interpretation of evolutionary dynamics.

In light of the foregoing, I maintain the opinion that this article should not be considered suitable for publication.

Response to Reviewer 2:

Reviewer comment 1:

Having carefully reviewed the most recent iteration of the manuscript titled "Dynamics of Reproductive Isolation in Stick Insects at the Transition Between Populations and Species," I must candidly express that my reservations remain unaddressed by the authors.

Response: We appreciated the reviewer again taking the time to evaluate our manuscript. We have now better addressed the reviewers' concerns and have further revised the manuscript as detailed below. In particular, we have added additional details on limitations of the work and better organized these in sections of 'Components of RI' and 'Limitations' in the Introduction and Discussion, respectively. This includes details on (1) estimating RI near the beginning of the manuscript when we introduce our experimental approach (see Components of RI, lines 143-175), (2) a second explicit section on limitations and future directions in the revised Discussion (lines 312-320) and (3) a revised elaboration of the implications of demographic history (lines 321-329). These main text sections now contain the logic and many of the citations used in our response letters. We hope that these additions satisfy the reviewer, and demonstrate our sincere attempts to produce a balanced manuscript with conclusions supported by the extensive data.

Reviewer comment 2:

The issue of experimental control persists without resolution. Mere citation of another article, wherein the relevant information is similarly obscure, falls short of constituting a control.

Response: With all due respect, the cited article, published in Nature Ecology and Evolution, provides details of morphological measurements not contributing strongly to RI and where they stem from, and how they were related to divergence time in a time-calibrated phylogeny.

We note, however, that the issue of control traits is complicated. Controls make sense in many areas of science of course, and our use, for example, of within taxon/population pairs for mating experiments serves as a control for between taxon/population pairs that allows us to infer mating preference (rather than a general propensity to mate, lines 475-493). In contrast, there is nothing to control for when estimating the rate at which the components of RI evolve (the rate is the rate), and thus including control traits in this sense is not meaningful. It would of course be different if we were asking whether these components of RI evolve faster/slower than random traits, but that is not a question posed or addressed in the manuscript. In a sincere attempt to satisfy Reviewer 2, we have (somewhat reluctantly as we think it doesn't really add to the manuscript) added reference to morphological traits that provide some basis for how traits not involved in RI might evolve. We make reference to this in the revised manuscript (see lines 165-175). But again, this is not meant as a 'control' in a strict sense as we are not testing the hypothesis that components of RI in general exhibit different dynamics than other traits.

Rather, we are (most importantly) estimating the dynamics for each component of RI , comparing these two components of RI to each other, and (again somewhat reluctantly) comparing these two to the specific morphological traits considered in past work. We have made this more clear in the revised manuscript (lines 165-175).

We have also clarified the purpose of the morphological data and the goals of the paper (the hypotheses we are versus are not testing) in the revised Introduction in the Components of RI subsection:

'We have previously shown that a suite of morphological traits--including body size and shape--diverge through time in a linear fashion (Riesch et al. 2017). These morphological traits do not clearly generate RI and thus act as a type of *a priori* expectation for evolutionary dynamics of traits that might be less critical for speciation. This allows us to here ask whether habitat and sexual isolation also exhibit linear dynamics similar to the morphological traits, or whether instead these components of RI exhibit different (i.e., non-linear) evolutionary dynamics. Linear dynamics would not imply that habitat and sexual isolation are unimportant for speciation, as, after all, the evolution of RI is central to it under many verbal and formal models speciation (Mayr 1942, Coyne & Orr 2004). Moreover, we emphasize that we do not intend this as a test of the null hypothesis that components of RI exhibit different dynamics from other traits, as this is not the focus of this paper and would require a much larger suite of traits to generate a null distribution. Instead, we instead use past work on these morphological traits as a baseline expectation for components of RI.'

Finally, we reiterate that any component of RI that evolves before RI is complete (the case for all the within-species comparisons in our revised manuscript) contributes at least somewhat to speciation (as noted in the revised manuscript, lines 192-198 and 318-320)

Reviewer comment 3:

The question pursued by the authors is undeniably intriguing and holds paramount significance within the field of speciation biology - the comprehension of the evolutionary dynamics underpinning reproductive isolation. Nevertheless, my skepticism regarding the appropriateness of relabeling isolated traits as "reproductive isolation" remains unabated. This semantic shift essentially transforms the conducted experiments into an inquiry into reproductive isolation, a conceptual realignment that I consider fundamentally misguided. The assertion that other authors employ a similar approach lacks compelling substantiation. Reproductive isolation encompasses more than the two traits under scrutiny; genomes abound with various incompatibilities contributing to reproductive isolation.

Response: We are pleased to hear that the topics we address are deemed of general significance and we fully acknowledge the reviewers point that many different components of RI and types of incompatibilities may contribute to speciation. This point was stressed by us in previous versions of the manuscript and is now very clearly acknowledged in the revised manuscript (lines 30-31, 82-88, 143-144, 159-164, 192-198

and 312-320). Our initial submission outlined why it is likely that host and mate preference generate components of reproductive isolation (RI). We added text bolstering the evidence and logic for this in the two revisions to date. During all rounds of revision, we have toned down our language extensively. Nonetheless, due to persistent concerns of the remaining reviewer on the topic of 'RI', we have taken the suggestion to explain how our data can be viewed as 'proxies' for components of RI at their first mention, specifically as these are experimental measures of these components of RI, not RI as it actually operates under natural conditions. Moreover, we have carefully gone through and revised the text throughout the manuscript to ensure that we refer to these as components of RI, not simply RI, which could be interpreted as total RI (something we do not wish to imply). As part of this effort, we have changed the title to include "sexual and habitat isolation" rather than just "reproductive isolation". Finally, we emphasize that although additional traits and components of RI can always be measured, the two we studied are commonly the focus of speciation research and our data itself represent a massive empirical effort with many thousands of host and mate preference trials between many pairs of taxa spanning the speciation continuum (resulting in arguably one of the most systematic evaluations to date of the evolution of these components of RI). Finally, any component of RI that evolves before RI is complete (the case for all the within-species comparisons in our revised manuscript) contributes at least somewhat to speciation. These points are made in the Introduction (lines 143-175) and again in a new section on limitations in the Discussion (lines 312-320). For full details see the new subsection of our revised manuscript on limitations of our experimental approach, with a few particularly relevant details pasted below:

From the Introduction:

'We note that we experimentally estimate two components of RI, not total RI. Moreover, our estimates come from experiments that do not fully recreate natural conditions (this is of course true of nearly all experimental studies of extrinsic RI). Thus, our experimental estimates of habitat and sexual isolation constitute proxies for how these components of RI would operate in nature. We nonetheless refer to these metrics as "components of RI" hereafter for simplicity and to be consistent with common usage in the literature (e.g., Orr 1995, Coyne & Orr 2004, Funk et al. 2006, Nosil 2012).'

'As noted above, we acknowledge that we here studied two components of RI (habitat and sexual isolation), among many possible ones. Future studies could usefully evaluate additional components of RI, including intrinsic RI, to better understand the contribution of each to speciation. Nonetheless, as detailed above, we note that the two particular components of RI studied here are likely relevant for understanding speciation in *Timema* and mate preference is the component of RI studied in most classic studies of RI versus genetic distance (Coyne & Orr 1989, Mendelson 2003, Presgraves 2002, Price & Bouvier 2002). Moreover, dynamics for these components of RI are compared to those for morphological traits thought to be less critical for speciation.'

From the Discussion:

'As acknowledged above, we studied two components of RI (habitat and sexual isolation), among many possible ones. Evolutionary dynamics for other components of RI, or even the same components of RI measured under different conditions, could exhibit different dynamics, including dynamics that differ from traits not associated with RI to different extents. Future work could fruitfully examine additional components of RI, especially intrinsic RI. Nonetheless, the two components studied here are very likely to generate meaningful RI in our study system and thus to be relevant for understanding speciation. This is especially likely as our demographic models suggest ongoing gene flow, which means that RI is incomplete between many taxon pairs and thus all components can make a contribution to the total RI.

'.

Reviewer comment 4:

Within this framework, wherein the quantified traits represent but an enigmatic facet of reproductive isolation, the absence of a discernible correlation between the "divergence of these traits" and "genetic divergence" does not constitute a result worthy of publication. Do these traits contribute to reproductive isolation with uniform potency across all pairs? Did these traits evolve before or after the emergence of other barriers for all pairs?

Response: Please see our response to points above in terms of new sections of our revised manuscript dedicated to limitations of our work, including the estimation of RI. However, with all due respect we argue that this particular claim of the reviewer fundamentally misrepresents our core findings:

We do not report the absence of a discernible correlation. Rather, we report correlations between divergence in our studied components of RI and genetic divergence, for both components. The nuance is that one correlation is linear (mating isolation) and the other is non-linear (habitat isolation). These results contribute to understanding of how different traits and components of RI evolve during the extended speciation process, and we thus argue are worthy of publication. To try and clarify this point of confusion we have carefully checked and revised our wording throughout the further revised manuscript.

Reviewer comment 5:

The discussion concerning demographic history remains conspicuously lacking in depth. Despite the execution of demographic analyses, the findings present a confluence of diverse scenarios, some involving isolation and others not, further complicating the interpretation of evolutionary dynamics.

Response: We appreciate that demographic history can be complex, and in fact it was the reviewers comments during an earlier round of review that promoted us to embrace this complexity via the addition of demographic analyses (which in the previous round of

review the reviewer did acknowledge were an impressive addition). To address the issue raised here by the reviewer we have expanded our discussion of demographic history and its consequences in the Limitations subsection of revised manuscript (lines 321-329). We now clarify what insights can be made from our data - the relationships observed between components of RI and genetic divergence are straightforward - but interpreting genetic divergence itself is more nuanced.

Reviewer comment 6:

In light of the foregoing, I maintain the opinion that this article should not be considered suitable for publication.

Response: We hope that the further revisions we have described above, in the context of comments from the reviewer, have improved the manuscript sufficiently that it is now deemed suitable for publication.